# Biosynthesized gold nanoparticles that activate Toll-like receptors and elicit localized light-converting hyperthermia for pleiotropic tumor immunoregulation

Hao Qin[1,5], Yang Chen[1,2,5], Zeming Wang[1,2], Nan Li[1,2], Qing Sun[1,2], Yixuan Lin[1,2], Wenyi Qiu[1,2], Yuting Qin[1], Long Chen[1], Hanqing Chen[3], Yiye Li ®[1], Jian Shi[1], Guangjun Nie ®[1,2,4] ✉ & Ruifang Zhao ®[1,4] ✉

Manipulating the tumor immune contexture towards a more active state can result in better therapeutic outcomes. Here we describe an easily accessible bacterial biomineralization-generated immunomodulator, which we name Ausome (Au + [exo]some). Ausome comprises a gold nanoparticle core covered by bacterial components; the former affords an inducible hyperthermia effect, while the latter mobilizes diverse immune responses. Multiple pattern recognition receptors actively participate in Ausome-initiated immune responses, which lead to the release of a broad spectrum of pro-inflammatory cytokines and the activation of effector immune cells. Upon laser irradiation, tumor-accumulated Ausome elicits a hyperthermic response, which improves tissue blood perfusion and contributes to enhanced infiltration of immunostimulatory modules, including cytokines and effector lymphocytes. This immune-modulating strategy mediated by Ausome ultimately brings about a comprehensive immune reaction and selectively amplifies the effects of local antitumor immunity, enhancing the efficacy of well-established chemo- or immuno-therapies in preclinical cancer models in female mice.

Cancer immunotherapy aims to leverage the potential of the immune system to combat tumors[1–3]. However, the immune system is capable of either constraining tumor progression or accelerating tumor development, dependent on tumor microenvironmental conditions[4–6]. The immunosuppressive feature of the tumor microenvironment (TME) can weaken the tumor cell surveillance activity of the immune system as well as the antitumor immune response induced by cancer immunotherapeutics[7,8]. Therefore, manipulating the immune state within the TME has been identified as a key feature in the design of an

effective cancer therapeutic strategy. Although various interventions, including bacteria-derived agonists (e.g., MPLA, poly-ICLC)[9,10], cytokines (e.g., interleukin-2, interferon-α)[11], immune checkpoint blockade antibodies [e.g., antibodies against programmed cell death protein 1(PD-1) or its ligand PD-L1][12], and vascular remodeling agents [e.g., antibodies vascular endothelial growth factor (VEGF) or angiopoietin 2 (ANG2)][13], have been developed and explored to regulate the TME to augment immunity, the therapeutic outcomes remain unsatisfactory. The main reason for the ineffectiveness of these agents is that a single

[1]CAS Key Laboratory for Biomedical Effects of Nanomaterials and Nanosafety, CAS Center of Excellence in Nanoscience, National Center for Nanoscience and Technology, Beijing 100190, P. R. China. [2]Center of Materials Science and Optoelectronics Engineering, University of Chinese Academy of Sciences, Beijing 100049, P. R. China. [3]Beijing Key Laboratory of Environmental Toxicology, Department of Toxicology and Sanitary Chemistry, School of Public Health, Capital Medical University, Beijing 100069, China. [4]GBA National Institute for Nanotechnology Innovation, Guangdong 510700, P. R. China. [5]These authors contributed equally: Hao Qin, Yang Chen. ✉e-mail: niegj@nanoctr.cn; zhaorf@nanoctr.cn

target or pathway may not be sufficient to fully mobilize immune cell function in the tumor microenvironment[14–16]. Moreover, systemic administration of these regulating factors often leads to severe off-target toxicity, thus restricting the use of optimal doses and further reducing treatment efficacy[17]. Hence, an ideal immunomodulator is to mobilize diverse immune responses and selectively functions at the tumor site, thus reaching augmented immunity and enhanced antitumor efficacy.

In this work, we describe an *Escherichia coli* (*E. coli*) biomineralization-generated gold nanoparticle (Ausome) containing numerous immune agonists and controllable hyperthermia induction to realize efficient immunomodulation at the targeted tumor site. Ausome is produced via the intracellular reduction of ionic gold into solid gold nanoparticles, which are then secreted to the extracellular environment with bacterial components coating the surface. These compositions endow Ausome with the intrinsic immune stimulatory ability and photothermal conversion capacity. Based on these features, we propose a multi-layered modulation when applying Ausome to regulate the tumor immune microenvironment, as shown in Fig. 1. The bacterial molecules stimulate multiple pattern recognition receptors, leading to the extensive release of immune-promoting cytokines and activation of effector lymphatic cells, including T cells and natural killer (NK) cells; the gold nanoparticle inner core, with intrinsic photothermal conversation properties, can mediate a light-controlled, local, mild hyperthermia, which promotes blood perfusion and contributes to the infiltration of effector lymphatic cells, thus selectively amplifying the immune response in the tumor area. Such a pleiotropic modulation pattern elicits enhanced antitumor immunity without additional doses of the material, consequently eliminating the safety concerns caused by the systemic administration of immunomodulators. Ausome-generated immune regulation also improves the sensitivity to and therapeutic efficacy of other well-established tumor therapies, such as chemotherapy and immune checkpoint blockade. Overall, the present study introduces a biomineralization process capable of generating a potent immunomodulating nanomaterial, biosynthesized in an eco-friendly and cost-effective process. The product exhibits potent TME manipulation capacity with multi-layered regulatory modes, and holds great potential for diverse therapeutic applications.

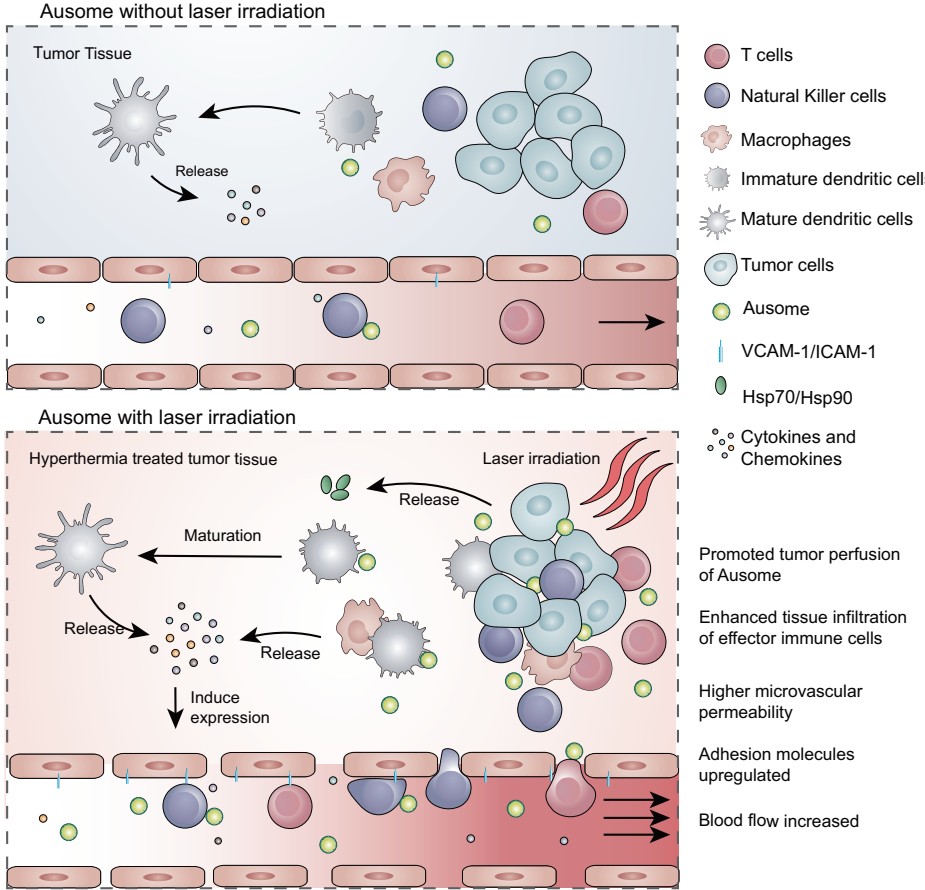

**Fig. 1 | Schematic illustration of tumor microenvironment manipulation by Ausome-mediated, multi-layered modulation.** Ausome comprises a bacterial component shell around a gold nanoparticle inner core. After intravenous injection, the diverse microorganism-derived danger signals stimulate a systemic immune response, including pro-inflammatory cytokine release, immune cell activation and expansion. Next, the tumor-accumulated Ausome is exposed to laser irradiation to heat the tumor tissue and generate a mild hyperthermia, which facilitates the intratumoral infiltration of additional effector immune cells by increasing the blood flow, promoting the permeability of blood vessels and upregulating adhesion molecules on vascular endothelial cells. The hyperthermia-induced blood perfusion also enhances the leakage of circulating Ausome into the tumor tissue, which can further enhance the immune reactions in the tumor region. This multi-layered modulation strategy circumvents the overactive immune reactions elicited by the high systemic doses of immunomodulators normally required for effective treatment, and selectively amplifies immune recovery in the tumor region to enhance the potency of immune surveillance of tumor cells.

## Results

### Biosynthesis of Ausome

We chose the DH5α strain of *Escherichia coli* (*E. coli*) as the bio-factory to generate gold nanoparticles from chloroauric acid (HAuCl₄). The bacteria were cultured in LB (Luria-Bertani) medium and grown to the logarithmic phase, followed by isolation and dispersion in phosphate buffer saline (PBS) solution containing 1 mM HAuCl₄. The solution turned pink after incubation for 8 h, and then gradually darkened with extended incubation time (Fig. 2a), implying the continuous formation of gold nanoparticles with time. To monitor the synthetic process, we isolated the bacteria from the reaction solution and imaged them by transmission electron microscopy (TEM) and scanning electron microscopy (SEM) at different time points (Fig. 2b, c). Prior to interaction with ionic gold, the *E. coli* were structurally complete, showing a smooth surface. A clear capsule layer and fimbriae structure were apparent in the periphery of the bacterial cells. After incubation for 0.5 h, the capsule layer of the *E. coli* became vague and ultrasmall nanoparticles emerged in the extracellular polymeric substance (EPS). This phenomenon, occurring in the first few hours of incubation, is consistent with the bacterial mechanism of protection from environmental heavy metal stress[18]. Au³⁺ was absorbed into the EPS and reduced by reductants, including enzymes, embedded within EPS[18]. Intracellular gold nanoparticles with diameters of approximately 30 nm were first observed at 4 h of incubation, followed by the alignment of the nanoparticles at the submarginal region of the bacteria at 12 h. After 48 h, large amounts of nanoparticles accumulated at the submarginal area and tended to be secreted from the bacterial body, scattering in the extracellular area (Fig. 2b, c), indicating the release of gold nanoparticles from the bacterial cells, after intracellular biosynthesis.

We observed a damaged capsule layer in the bacteria after a few hours' incubation (Fig. 2b, c), which may contribute to the diffusion of Au³⁺ into cells, followed by generation of the gold nanoparticles. To explore the exact compartment in which Au³⁺ reduction occurs, we incubated the *E. coli* with HAuCl₄ for 48 h, and then generated ultrathin sections in which the metal could be visualized. Gold nanoparticles were found at the margin of the bacterial cells, specifically in the periplasmic space between the outer and cytoplasmic membranes (Fig. 2d), where Au³⁺ reduction has been reported[19]. The embedding of 30-nm particles in the periplasmic space led to bacterial membrane swelling (Fig. 2d); this localized pressure altered the curvature of the outer membrane, which has been identified as a mechanism stimulating bacterial membrane vesicle biogenesis[20]. These observations suggest that the intracellularly formed gold nanoparticles may be excreted to the extracellular space in a membrane-bubbling manner. SEM and TEM images of the bacteria further support this hypothesis that nanoparticles located on the surface of cells created bulges underneath a thin layer of the membrane (Fig. 2e and Supplementary Fig. 1), indicating vesicle formation before nanoparticle release. We termed the biosynthesized and secreted gold nanoparticle Ausome after its gold (Au) core and bacterial membrane coating.

We then used a two-step centrifugation procedure to isolate Ausome from the culture medium. First, bacteria bodies were sedimented by low-speed centrifugation (5000× *g*), and discarded; the supernatant was subsequently centrifuged at a high speed to separate Ausome (in the pellet) from proteins, unreacted Au³⁺ and membrane vesicles devoid of gold nanoparticles (in the supernatant). To obtain Ausome with optimal quality, two different centrifugation forces in the second step were explored. A relatively low speed (11,000× *g*) sedimented Ausome with more uniform size (30–40 nm in diameter), while higher speeds (19,000× *g*) generated both large (30–40 nm in diameter) and additional, smaller nanoparticles (10 nm in diameter; Supplementary Fig. 2). In consequence, we chose the lower centrifugation force (11,000× *g*) at the second centrifugation procedure during Ausome preparation. TEM images of the final products suggest

that, the two-centrifugation-step protocol removed most, if not all, impurities from Ausome (Supplementary Fig. 2).

To examine the time-dependence of Ausome generation, we collected the gold nanoparticles after a series of incubation times, followed by UV-visible absorption spectroscopy measurement. A significant enhancement of absorbance in the first 48 h revealed the rapid generation of Ausome, followed by a steadily slow increase over the next 8 days (Fig. 2f). We further confirmed this process by quantification of Au by inductively coupled plasma mass spectrometry (ICP-MS; Fig. 2g). Again, the *E. coli*-mediated biosynthesis of Ausome was composed of a fast growth phase during the initial 48 h, followed by a slow growth phase for more than 8 days. Together, these procedures for Ausome generation, including the intracellular reduction of Au³⁺, membrane bubbling-mediated secretion and isolation, are summarized in Fig. 2h, which demonstrates the facile and eco-friendly approach to prepare Ausome.

### Characterization of the composition and activity of multifunctional Ausome

We next analyzed the structure and composition of Ausome, as well as its physicochemical properties and biologic activities. TEM imaging and dynamic light scattering (DLS) analysis revealed that Ausome nanoparticles were mostly spherical and well dispersed (Fig. 3a). The higher magnification TEM images of Ausome show an apparent core-shell structure with a high-contrast inner core and a low-contrast surface coating with a thickness of approximately 3.19 nm (Fig. 3b). High-resolution images display multiple lattice planes inside Ausome, with fringe spacings of approximately 0.204 nm and 0.235 nm, which corresponds to interplanar distances of d (111) and d (200) in nanocrystalline gold (Fig. 3c). Selected area electron diffraction (SAED) of Ausome revealed powder rings consistent to (111), (200), (220) and (311) reflections of nanocrystalline gold (Fig. 3d), confirming that the inner core component of Ausome was gold crystal. Thermogravimetric (TG) analysis revealed that the gold inner core of Ausome was approximately 74% by weight, while the remaining 26% (including ~6% crystal water) were heat-labile and lost during the heating process (Fig. 3e). Elementary mapping showed N, S and P in Ausome, suggesting that the heat-labile molecules may include proteins and phospholipids (Supplementary Fig. 3).

To identify the exact composition of biological components in Ausome, we performed both lipidomic and proteomic analyses. A high content of phospholipids and phosphorus-free lipids, which are the main building blocks of natural lipid bilayers[21], were identified via either positive electrospray ionization or negative electrospray ionization (Fig. 3f, Supplementary Table 1). The lipidomic results indicate the presence of a membrane structure in Ausome. The major lipids comprising *E. coli* membranes, including phosphatidylethanolamine (PE), phosphatidylglycerol (PG) and cardiolipin (CL)[22], were decreased in Ausome, while other lipids that are not common in *E. coli*, such as glycerophospholipids (PC), sphingolipids (SM), diacylglycerol (DG) and phosphatidic acid (PA), were in high abundance (Fig. 3f, Supplementary Table 1). The unusual lipid composition was likely related to bacteria autophagy and membrane rearrangement, potentially due to the nutrition-deficient culture conditions and membrane vesicles formation[21,23–25]. Lipid A, the lipid portion of lipopolysaccharide (LPS), functions as an anchor to fix LPS on the outer membrane of bacteria[26], was detected using positive ion mode (Fig. 3f, Supplementary Table 1). This observation indicates the presence of outer membrane components on the surface layer of Ausome. Together these results further support our hypothesis that *E. coli*-biosynthesized gold nanoparticles were secreted along with membrane vesicle genesis and surface coating with bacterial membrane components. We next performed proteomic analysis to identify the protein composition of Ausome. In addition to the proteins located in the cytoplasm, membrane resident proteins, especially those located in the outer membrane, were detected (Fig. 3g,

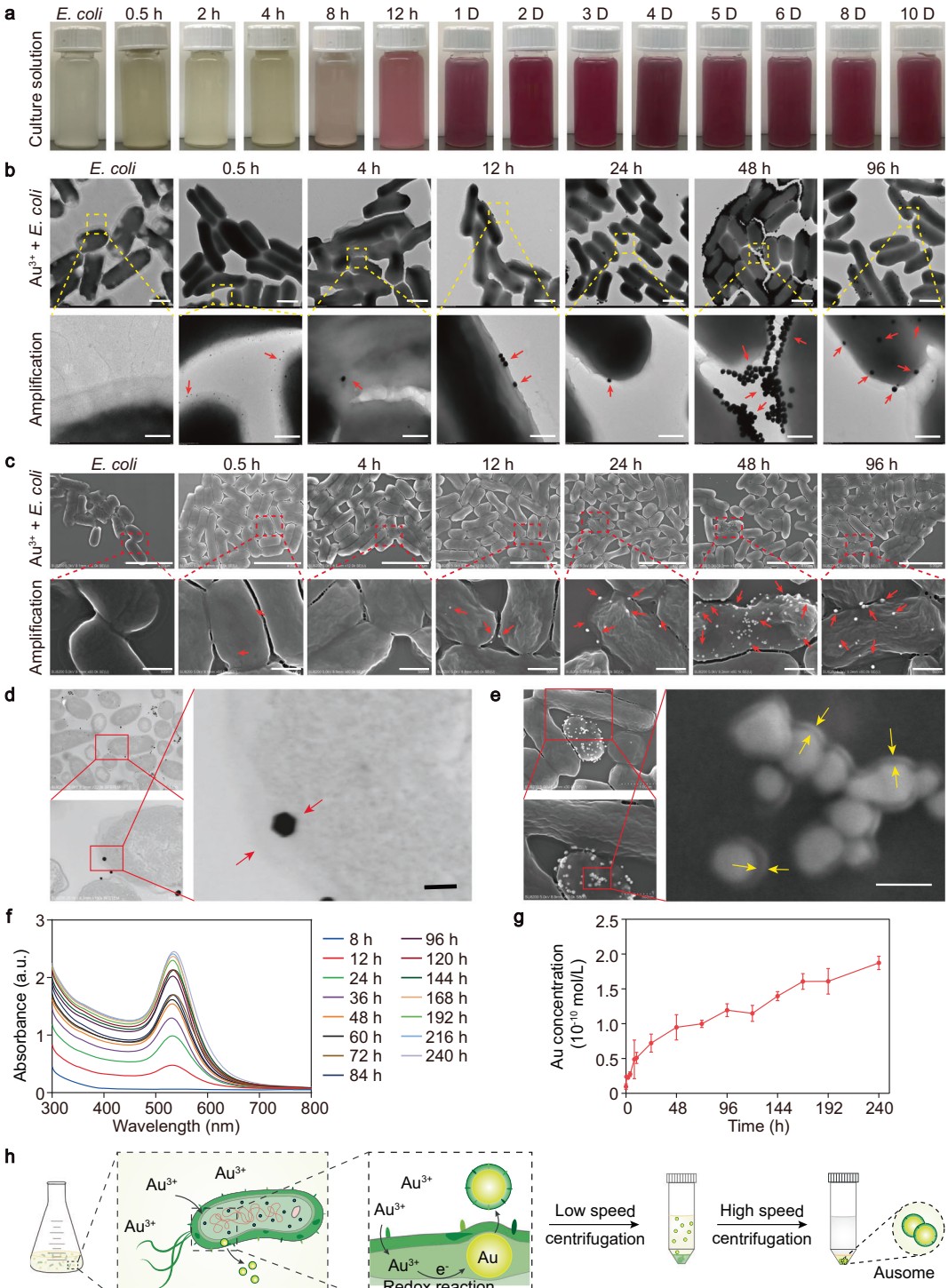

**Fig. 2 | Ausome production by *E. coli*. a** *E. coli* cultured in the presence of HAuCl₄ for the indicated times (h, hours; D, days). **b,c** Transmission electron microscopy (TEM, **b**) and scanning electron microscopy (SEM, **c**) images of *E. coli* and the generated gold nanoparticles at the indicated culture time points. Dashed boxes indicate the enlarged region. The red arrows indicate the biomineralized gold nanoparticles. Scale bars: 1 μm (**b**, upper panels), 200 nm (**b**, lower panels, 4 μm (**c**, upper panels), 500 nm (**c**, lower panels). **d** Ultrathin sections of *E. coli* after incubation with HAuCl₄ for 48 h imaging using scanning transmission electron microscope (STEM). Red arrows indicate the bacterial compartment of Ausome

generation. Scale bars, 50 nm. **e** SEM images of Ausome present on the surface of *E. coli* after 48 h incubation. Yellow arrows indicate the thin coating on Ausome. Scale bar, 50 nm. **f, g** UV-visible absorption spectra (**f**) and ICP-MS (**g**) quantification (*n* = 3 biologically independent samples) of secreted Ausome isolated from *E. coli* after the indicated culture times. **h** Schematic illustration of the generation and isolation of bacteria-generated Ausome. The numerical data in (**g**) are presented as the mean ± standard deviation (s.d.). These experiments (**a**–**f**) were repeated three times independently with similar results. Source data are provided as a Source Data file.

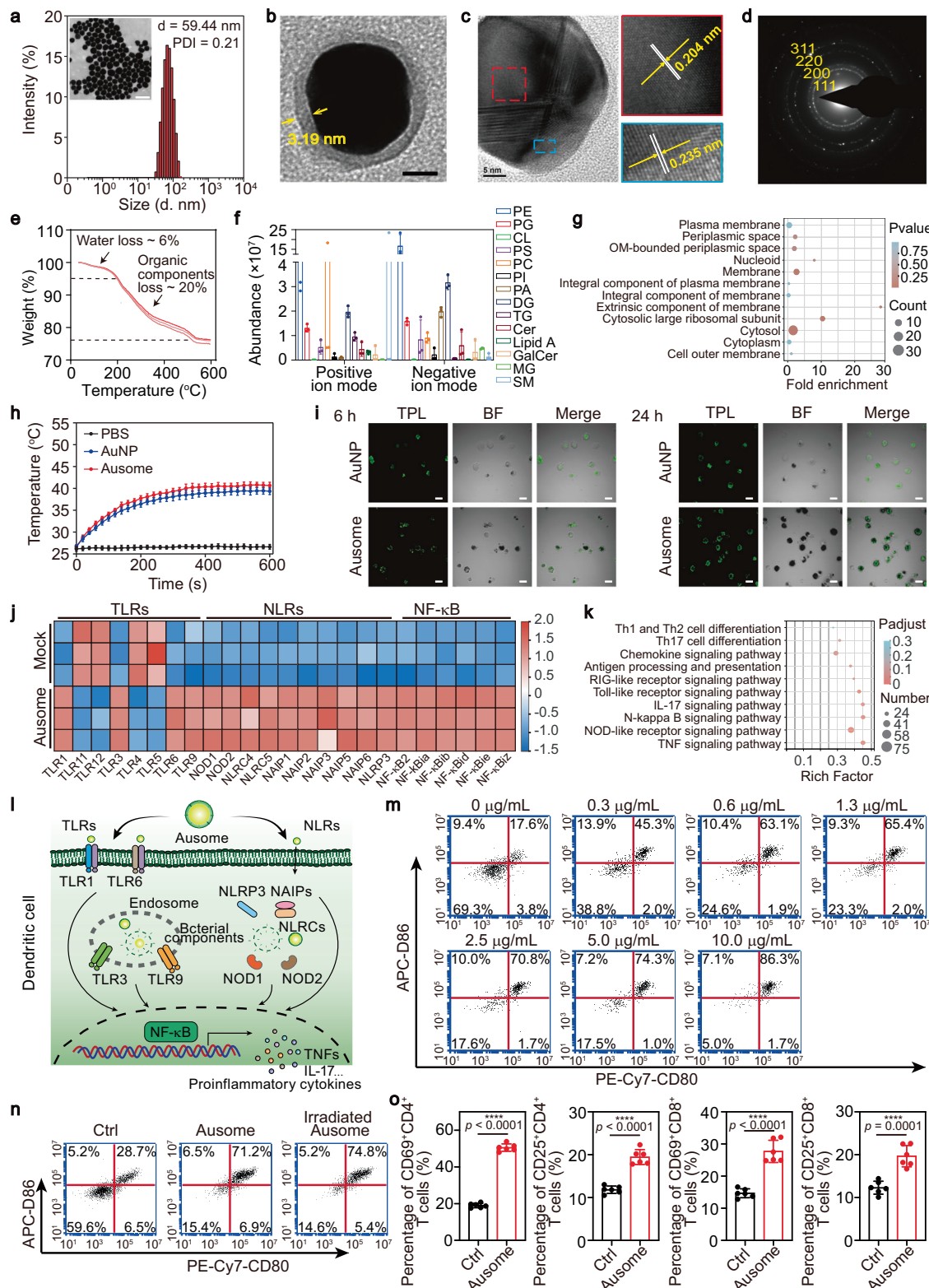

Supplementary Table 2). The translocation of the inner membrane protein, SecG, and the outer membrane protein assembly machinery, BamA, has been used to distinguish between the inner and outer membranes[27,28]. The presence of BamA, and the absence of SecG, demonstrated that the Ausome surface coating comprised mainly outer membrane components (Supplementary Table 2).

We next investigated the physicochemical properties and biological activities of Ausome. Gold-based nanomaterials exhibit localized

plasmon surface resonance and have been thoroughly studied as photothermal agents[29]. We therefore verified and evaluated the photothermal conversion ability of Ausome. To guarantee efficient light absorption, as well as sufficient tissue penetration depth, we chose a laser wavelength of 660 nm to irradiate the nanomaterial. Solutions of different Ausome concentrations were exposed to the laser at the same power density (1.5 W/cm²) and the temperature was monitored

**Fig. 3 | Characterization of Ausome: structure, composition, physicochemical properties and biological activities. a** TEM image and hydrodynamic size of Ausome. Scale bar, 50 nm. **b** Enlarged TEM image of Ausome. Yellow arrows indicate the outer layer of Ausome. Scale bar, 10 nm. **c** High-resolution TEM images of Ausome. Dashed boxes indicate the different lattice planes, white bars and yellow arrows indicate the interplanar distances. Scale bar, 5 nm. **d** SAED pattern of Ausome. Yellow numbers represent the Miller index consistent with the powder rings. **e** TG analysis of lyophilized Ausome ($n = 3$ biologically independent samples). **f** Lipidomics analysis; histogram of the most abundant lipids in Ausome ($n = 3$ biologically independent samples). **g** Proteomics analysis of the protein composition in Ausome; bubble plot representation of enrichment analysis of ontology (GO) cellular components. **h** Temperature changes of AuNP and Ausome suspensions ($n = 3$ biologically independent samples) under laser irradiation. **i** Two-photon luminescence (TPL) images of BMDCs after incubation with AuNP or Ausome. Scale bars, 20 μm. **j**, Heat map of differentially expressed genes in BMDCs after treatment with Ausome ($n = 3$ biologically independent samples). **k** Bubble plot representation of KEGG enrichment analysis for the differentially expressed genes involved in immune modulation-corelated signaling pathways. **l**, Comprehensive schematic illustration of the mechanisms and effects of Ausome-elicited immune responses. **m, n** Representative flow cytometry dot plots of CD80+ and CD86+ BMDCs, after treatment with Ausome for 24 h at the indicated doses (**m**) or with irradiated Ausome (**n**). **o** Frequencies of CD69 or CD25 highly expressed CD4+ and CD8+ T cells after incubation with Ausome activated BMDCs for 24 h and detected using flow cytometry ($n = 6$ biologically independent samples). The numerical data in (**f, h, o**) are presented as the mean ± s.d. ***$p < 0.001$, ****$p < 0.0001$; A two-tailed unpaired $t$-test (**o**) or two-tailed t-test with the Benjamini-Hochberg method to correct $P$ values (**g, k**) was used for statistical significance analysis. These experiments (**a–d, i, m, n**) were repeated three times independently with similar results. Source data are provided as a Source Data file.

using an infrared thermal camera. To generate a control, we chemically synthesized surface PEGylated gold nanoparticles (AuNP) with a similar size and absorption spectrum as Ausome (Supplementary Fig. 4a, b). Under irradiation, both the Ausome and AuNP suspensions exhibited rapid temperature increases in the first few minutes, with the maximum temperature concentration and laser power-dependent (Fig. 3h, Supplementary Fig. 4c–f), indicating that Ausome and AuNP can convert light energy into heat.

Since chemically synthesized gold nanoparticles tend to aggregate under freezing conditions but cryogenic storage is necessary to conserve the bioactivity of Ausome, therefore we examined the stability of Ausome after repeated freeze-thaw cycles or long-term freezing at −80 °C. No sediment was apparent in the thawed solutions and the TEM images exhibit monodispersed nanoparticles. DLS detected sizes and surface charges consistent with the fresh prepared Ausome further confirmed its stability under these storage conditions (Supplementary Fig. 5a–c). We also evaluated the influence of laser irradiation on the physiochemical features of Ausome. After exposing to laser for 30 min, Ausome was assessed using TEM and DLS, and little difference in morphology, diameter and zeta potential was observed when compared to fresh prepared Ausome (Supplementary Fig. 5a–c), revealing high laser stability of Ausome.

To assess the biological effects elicited by Ausome, we evaluated the interaction between Ausome and innate immune cells. After incubation with Ausome or AuNP for 6 or 24 h, we imaged bone-marrow dendritic cells (BMDCs) using two-photon laser scanning confocal microscopy; the two-photon luminescence (TPL) emitted from the gold nanoparticles was imaged to assess their cellular uptake behavior. Ausome-treated BMDCs showed strong intracellular TPL signals, which were enhanced with extended incubation time (Fig. 3i). We also evaluated cellular uptake by assessing the gold contents in cells by ICP-MS detection after the incubation step. More than 70% of the total Ausome was present associated with cells after 24 h (Supplementary Fig. 6a). In comparison, only 10% of the AuNP was taken up by BMDCs and there was no further accumulation with prolonged incubation time (Fig. 3i, Supplementary Fig. 6a). We also monitored the intracellular localization of Ausome by labeling the surface membrane layer with fluorescent probes. Confocal microscopy imaging and flow cytometry analysis both confirmed the high affinity of BMDCs to Ausome, with approximately 100% of the cells exhibiting associated Ausome and most of the fluorescent signals found co-localized with lysosomal compartments (Supplementary Fig. 6b). We then investigated interactions between Ausome and macrophages, one of the critical components of the first line of immune systems and respond quickly to exogenous stimuli; and macrophages can also be a hitchhike for the in vivo transport of Ausome[30]. The bright field images exhibited a large amount of intracellular gold nanoparticles after incubating with Ausome for 12 h. At the same time, AuNPs treated bone-marrow macrophages (BMDMs) showed little uptake of the nanoparticles

(Supplementary Fig. 6c). ICP-MS further quantified the internalized Au, approximately 30% internalized Ausome and about 5% internalized AuNPs were measured, respectively (Supplementary Fig. 6d). These data demonstrate a highly efficient recognition of Ausome by innate immune cells, likely due to the interaction between pathogen-associated molecular patterns (PAMPs) present on the Ausome and pattern recognition receptors (PRRs) displayed by the dendritic cells (DCs) or macrophages.

To gain more insights into these interactions and explore the subsequent immune reactions, we examined the differentially expressed genes in Ausome-treated BMDCs through transcriptome sequencing. Highly altered gene expression was apparent in the cells treated with Ausome (Supplementary Fig. 7). We observed the upregulation of extensive PRRs, including toll-like receptors (TLR; e.g., TLR1, TLR3, TLR6, TLR9) and nucleotide-binding oligomerization domain (NOD)-like receptors (NLRs; e.g., NOD1, NOD2, NLRCs, NAIPs, NLRP3), as well as transcription factor nuclear factor-kappa B (NF-κB; Fig. 3j), suggesting an immunomodulatory mechanism of simultaneous activation of diversified targets and the initiation of immune signaling pathways (including NF-κB). We next performed KEGG (Kyoto encyclopedia of genes and genomes) pathway analysis to investigate the functions of these differentially expressed genes. The most enriched genes were involved in pathogen molecule recognition, effector lymphocyte activation, immune signal transcription, pro-inflammatory cytokine regulation, and antigen processing and presentation (Fig. 3k, Supplementary Fig. 8a), indicating a comprehensive activation of innate and adaptive immune responses. As consequence, there was an upregulation of a broad spectrum of cytokines and chemokines (Supplementary Fig. 8b), which are important immune signals responsible for the conduction of downstream reactions. Together these data suggest that the abundant and diverse PAMPs on Ausome stimulate the activation of multiple PRRs to initiate a wide range of responses, including the upregulation of pro-inflammatory cytokines, differentiation of effector immune cell and activation of the antigen processing pathway (Fig. 3l).

After encountering danger signals, innate immune cells, such as DCs, respond to the stimuli immediately by maturing to initiate downstream immune reactions. In addition to secreting pro-inflammatory cytokines, DCs also upregulate the expression of co-stimulatory molecules, such as CD80 and CD86, which are used as biomarkers for immune activation efficacy[16]. We therefore used flow cytometry analysis to detect Ausome-triggered DC maturation. Nearly half of DCs appeared as mature, after treatment with Ausome at as low a dose as 0.3 μg/mL (for the purposes of our study, we define Ausome amount as Au content) for 24 h. Higher dose of Ausome triggered additional mature DCs, approximately 87% of them were CD80+CD86+ cells when incubation with 10 μg/mL Ausome (Fig. 3m). Shortening the exposure time to 6 h still remarkably elevated CD80+CD86+ BMDCs, while increasing the Ausome concentration further elevated the

proportion of matured DCs (Supplementary Fig. 9a, b). The rapid response of DCs revealed the high immunogenicity of Ausome, implying that the nanomaterial possesses the potential to trigger robust immune reactions. We also investigated whether laser irradiation would affect the immune-stimulating potency of Ausome. Ausome was exposed in laser (660 nm, 1.5 W/cm²) for 30 min and then added to the medium of BMDCs. Significantly enhanced frequency of CD80⁺CD86⁺ BMDCs was detected after incubation for 24 h and no apparent differences in triggering DCs activation were observed between Ausome with or without laser irradiation (Fig. 3n, Supplementary Fig. 9c, d), indicating irradiation affected little on immune-stimulating efficacy of Ausome. The synthetic AuNP or single laser irradiation treatment without any gold nanoparticles failed to facilitate the differentiation of BMDCs to mature phenotype (Supplementary Fig. 9c, d). Activated DCs then induced T cell responses either by surface co-stimulatory molecules and T-cell-receptor ligands or by secreted proinflammatory cytokines and growth factors, to modulate adaptive immune reactions[31]. Therefore, the activation of T cells by Ausome stimulated DCs was assessed. BMDCs were treated with Ausome for 12 h and then incubated with lymphocytes at a ratio of 1:5. After 24 h treatment, significantly increased proportions of CD4⁺ and CD8⁺ T cells with highly expressed CD69 or CD25 molecules were detected using flow cytometry (Fig. 3o). CD69 and CD25 are the hallmarks of T cell activation and clonal expansion[32], indicating the successful triggering of T cell responses by Ausome activated DCs. We next investigated whether Ausome could elicit direct cytotoxicity on tumor cells by incubating 4T1 breast cancer cells with Ausome. The cell viabilities were not affected by Ausome at concentrations up to 150 μg/mL, and increased dose or additional laser irradiation only elicited minor cytotoxicity (Supplementary Fig. 10a). No difference in cell viability was observed in 4T1 cells with other treatments, including laser only, AuNP or irradiated Ausome (single components or different status of Ausome mediated therapeutic process, Supplementary Fig. 10b). These results imply that direct cell killing may not be the key mechanism of Ausome mediated anti-tumor therapy in this study. Thus, Ausome generated from *E. coli* biomineralization manifests as a gold nanoparticle coated in bacterial membrane components, and exhibiting efficient photothermal conversion and strong immunogenicity.

## Evaluation of Ausome biosafety

In considering Ausome for in vivo application, the bacteria-derived molecules contained in the nanomaterial raise a safety concern[33]. We therefore performed a series of experiments to assess the biosafety of Ausome. We first tested for hemolysis after incubating Ausome with whole blood collected from mice; no lysis of red blood cells was apparent (Supplementary Fig. 11a). Apart from the blood cells, vascular endothelial cells are also exposed to Ausome and cytotoxicity of Ausome on vascular endothelial cells was evaluated then. Human umbilical vein endothelial cells (HUVECs) were incubated with Ausome at a concentration range of Ausome for 24 h, and no significant decrease in cell viability was observed, even in concentration as high as 300 μg/mL (Supplementary Fig. 11b). We also assessed the arrangement and structure of endothelial cells in vivo. High-dose Ausome (270 μg per mouse) were intravenously injected to Balb/c mice. The hearts, with adequate blood supply and abundant blood vessels, were isolated 48 h postinjection and immuno-stained to visualize vascular endothelial cells. No disturbance was observed in Ausome-treated blood vessels, endothelial cells maintained as narrow strip, which were closely connected together and arranged alongside the blood vessels, with intact structure and smooth surface (Supplementary Fig. 11c). Next, we assessed any Ausome-induced changes in cytokine production. In response to microbial molecules of Ausome, immune cells produce pro-inflammatory cytokines, including IFN-γ, TNF-α and IL-6. Although these cytokines are to be protective, the excessive

inflammatory response may cause tissue injury and even life-threatening cytokine storm[34,35]. To this end, we intravenously injected a series of Ausome doses, ranging from 5 to 20 mg/kg, into healthy mice. All treatments led to apparent elevations in the serum levels of IFN-γ, TNF-α and IL-6 (Supplementary Fig. 11d, e). We then repeatedly injected Ausome (three times, at 3-day intervals) into 4T1 tumor-bearing mice to investigate any severe tissue damage when applying the nanomaterial for tumor therapy (Supplementary Fig. 12a). No significant variations in blood biochemical indexes or obvious pathological changes were observed in animals from low-dose Ausome (15 mg/kg or less) treatment (Supplementary Fig. 12b, c), revealing no unrecoverable damages and severe systemic toxicity generated when Ausome applied at 15 mg/kg or less. While at increased Ausome dose (20 mg/kg), a significantly higher level of serum aspartate transaminase (AST) and infiltrated lymphocytes in the liver was observed in mice treated with 20 mg/mL Ausome, potentially due to liver dysfunction caused by the overactivated inflammation (Supplementary Fig. 12b, c). According to the above assessments, we chose a dosage of 15 mg/kg for in vivo usage of Ausome, based on safety concerns.

We then used female and male Sprague Dawley rats to validate the safety of Ausome when applied in vivo and to both female and male animals. A low dose of 6.5 mg/kg, and a high dose of 13 mg/kg, were tested through intravenous injection three times at a 3-day interval (Supplementary Fig. 13a). Compared to the non-treatment group, no significant difference in body weight was observed in both female and male rats during the experimental period (Supplementary Fig. 13b, c). Eight days after the third Ausome injection, the hematological parameters, serum biochemistry and histopathological estimation of major organs were measured. Increased amount of platelets was observed in Ausome-treated rats, both female and male, but within the normal range even in high-dose treatment[36] (Supplementary Table 3). For other parameters, including white blood cells count and ratios, amount and features of red blood cells, some minor fluctuations were found among the three groups in both female and male rats, but no statistic differences were observed (Supplementary Table 3). The biochemistry analysis showed little serum indicator levels fluctuation after Ausome treatment in both female and male rats (Supplementary Fig. 13d, e). The major organs, including the brain, heart, lung, liver, spleen, kidney, and intestine, were also isolated and evaluated for the safety profile of Ausome at the tested doses. We observed normal cell arrangements and detailed structures in all sectioned and stained organs, and no inflammatory or pathological changes were found (Supplementary Fig. 13f, g). In conclusion, Ausome elicited limited morphological and/or functional alterations in both female and male rats.

## Ausome-mediated local tumor hyperthermia and the consequent biophysical and biological effects

The characterization of Ausome revealed that the nanomaterial may be capable of achieving local tumor hyperthermia and provoking robust antitumor immunity with multi-layered immunomodulatory modes. To investigate whether Ausome's light-triggered thermal response can cause local hyperthermia and consequently affect tumors, we intravenously injected Ausome into 4T1 breast tumor-bearing Balb/c mice and then laser irradiated the tumor site, as illustrated in Fig. 4a. After intravenous injection into the tumor-bearing mice, we measured the time-dependence of tumor Ausome accumulation. The mice were imaged using an in vivo spectrum imaging system (IVIS) to visualize the behavior of Cy5.5-labeled Ausome at different time points. The Cy5.5 signal at the tumor site increased with treatment time, peaking at 6 h post-injection and then gradually decreasing. The fluorescent signal could still be observed at 72 h (Fig. 4b, c). To quantify the fluctuation of Ausome content over time, tumors harvested at different time points were digested and the amount of associated Au was determined by ICP-MS. The highest Au levels were present from 2–12 h

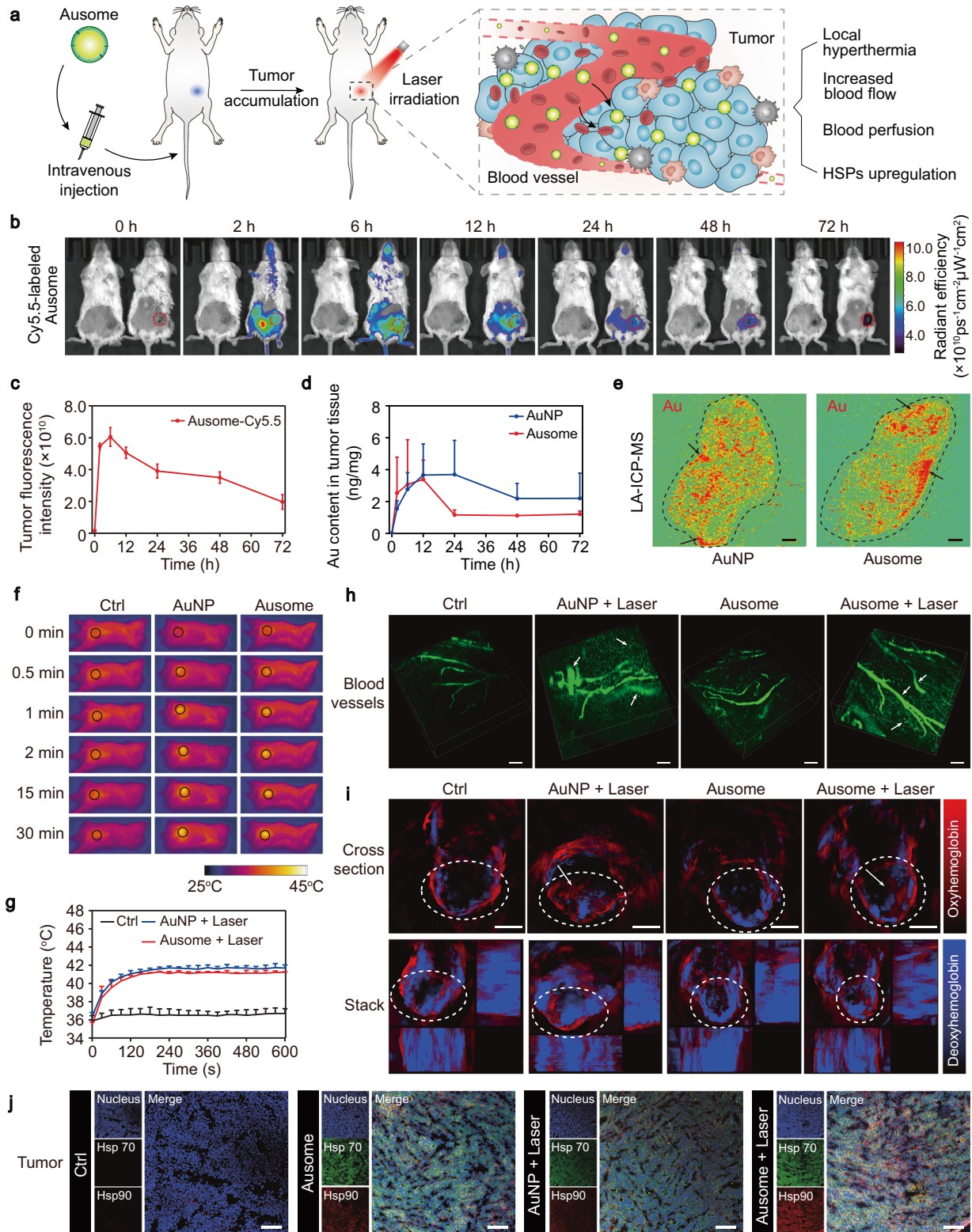

post-injection, after which time the levels of the metal decreased gradually (Fig. 4d). This observation was consistent with the results of the fluorescence labeling experiment. As a control, we also evaluated the tumor accumulation of AuNP. In contrast to Ausome, intravenously injected AuNP reached maximum accumulation at 12 h, which was sustained to 24 h, after which the content decreased but remained at

higher levels than that of Ausome for at least 72 h (Fig. 4d). We attribute this difference to the rapid elimination of the highly immunogenic Ausome by APCs in the tumor tissue. By laser ablation ICP-MS (LA-ICP-MS), we further investigated the intratumoral distribution of the nanoparticles at the time point in which they accumulated to the highest degree. As apparent in the images presented in Fig. 4e, there

**Fig. 4 | Ausome-mediated local tumor hyperthermia and the consequent physical and biological effects. a** Schematic illustration of the procedure to elicit local hyperthermia. **b, c** Representative IVIS images of 4T1 breast tumor-bearing Balb/c mice (**b**) and quantification of the fluorescence intensity at the tumor region (**c**) at the indicated time points post intravenous injection of Cy5.5-labeled Ausome. The mice on the right-hand sides of each panel are Ausome-treated, with untreated controls shown to the left (*n* = 3 mice). Red circles indicate the tumor region. **d** ICP-MS quantification of the Au content in tumors at the indicated times following intravenous administration of AuNP or Ausome (15 mg/kg) to tumor-bearing mice (*n* = 3 mice). **e** Laser ablation ICP-MS (LA-ICP-MS) images of Au distribution in tumors. The dashed line represents the tumor region and the dark arrows indicate areas with accumulated Au. Scale bars, 1 mm. **f, g** Representative infrared thermal images of 4T1 tumor-bearing mice (**f**) and the temperature changes at the tumor area (**g**) during laser irradiation (660 nm, 1.2 W/cm²; *n* = 3 mice). The black circles

indicate the tumor region. **h** 3D reconstructed confocal microscopy images of tumor blood vessels in mice treated with PBS, Ausome, AuNP with laser exposure or Ausome with laser exposure (660 nm, 1.2 W/cm², 30 min). The blood vessels were visualized by intravenous injection of FITC-labeled dextran (3 mg per mouse), immediately after irradiation. The white arrows indicate increased blood flow or leaked FITC-dextran. Scale bars, 200 μm. **i**, Living MSOT images of mice treated with PBS, Ausome, AuNP or Ausome-mediated local hyperthermia. The dashed white circles indicate the tumor zone, white arrows indicate deep blood signals in the tumor tissue. Scale bars, 5 mm. **j** Confocal microscopy images of tumor sections immunofluorescence-stained using antibodies against Hsp70 and Hsp90 (blue: nucleus, green: Hsp70, red: Hsp90). Scale bars, 200 μm. The numerical data in (**c**), (**d**) and (**g**) are presented as the mean ± s.d. These experiments (**e, h**–**j**) were repeated three times independently with similar results. Source data are provided as a Source Data file.

were regions with high signal density, mainly in the periphery of tissue rich in blood vessels, while dispersed Ausome or AuNP were also present in the inner areas of the tumor. These findings suggest that the nanoparticles are capable of reaching the tumor site via the blood-stream and then infiltrating to the inner zone of the tissue. We also evaluated the distribution of Ausome in other tissues. There were significant levels of high content of Au in the liver, spleen, lymph nodes, and lungs, i.e., either peripheral lymphatic tissues or organs with innate immune features, at 6 h after injection. We observed further accumulation in the liver, spleen and lungs at 24 h (Supplementary Fig. 14a, b), revealing a systemic immune response to the exogenous nanoparticles.

Next, we performed laser irradiation to generate Ausome-mediated local hyperthermia. The tumors were exposed to the laser (wavelength 660 nm; power density 1.2 W/cm²) at 6 h post Ausome injection (15 mg/kg), when the maximum amount of Ausome accumulates within tumors. The temperature variation within the tumors was monitored using an infrared thermal camera. The tumors heated up rapidly, plateauing in temperature at 2 min, with a maximum temperature of approximately 41 °C (Fig. 4f, g). We also monitored AuNP-induced tumor heating under laser irradiation. Mice administered the same amount of AuNP showed similar effects as Ausome treatment, while negligible temperature increases in tumors were observed in the PBS-treated control group (Fig. 4f, g). This mild temperature increases in the Au-containing tumors, which did not directly induce appreciable damage to the tumors (Supplementary Fig. 15), has been shown to accelerate blood flow, facilitate blood perfusion, and upregulate heat shock protein (HSP) expression[37–39]. Since these effects can contribute to the infiltration of therapeutic agents and/or potentiate immune reactions, hyperthermia has been widely utilized to locally amplify the efficiency of drugs, enhancing their intratumoral delivery and/or promoting the tumor immune state[40–43]. We therefore explored the blood flow in the tumors after 30 min of laser irradiation. To visualize the circulation, we intravenously injected fluorescein isothiocyanate (FITC)-labeled dextran and imaged the tumor using laser scanning confocal microscopy. 3D reconstructed images display thin vessels and dim FITC signals in untreated tumors, while both the Ausome- and AuNP-mediated thermal increases led to thick and bright fluorescent flow, demonstrating that the localized hyperthermia indeed resulted in increased tumoral blood perfusion (Fig. 4h). Numerous, dispersed FITC signals were also present in the extravascular region of the hyperthermic tumors, suggesting that heat promoted the extravasation of the blood components, which was confirmed by H&E staining of tumor tissue sections excised after laser exposure (Fig. 4h, Supplementary Fig. 16). By confocal microscopy images, we examined partial blood flow at a microscopic level, so we next investigated the blood perfusion macroscopically by real-time multispectral optoacoustic tomography (MSOT). The tumors were segmentally scanned to collect hemoglobin (Hb) and

oxyhemoglobin (HbO₂) signals, as indicators of blood flow and oxygenation. In untreated tumors, the Hb and HbO₂ signals appeared primarily in the tissue periphery, which is consistent with the absence of a blood supply in the deeper tumor regions (Fig. 4i). In comparison, markedly increased Hb and HbO₂ were detected both in the AuNP- and Ausome-mediated hyperthermic tumors, particularly in the interior regions (Fig. 4i), further demonstrating a markedly enhanced blood perfusion after the elevation in temperature. Except for the physical effects, heat can also trigger biological reactions, with HSPs activation as the initial responders prior to initiation of the downstream pathways[38,39]. Therefore, we examined the expression of HSP70 and HSP90 in tumors to further assess the hyperthermia induced effects. Immunofluorescence images show an elevated signal intensity within the tumors treated with AuNP-mediated thermal effects (Fig. 4j). In comparison to treatment with Ausome alone, the inclusion of laser irradiation resulted in greater levels of HSP expression (Fig. 4j). These findings verify that Ausome is capable of mediating light induced hyperthermia in tumor area and trigger heat related biophysical or biological effects, which implies its potential to controllably enhance immune regulation effects.

## Ausome-mediated multi-layered modulation for amplified antitumor immunity

To assess Ausome-mediated immune regulation, we examined the cytokine profiles and immune cell composition in tumor tissues. As shown in Fig. 5a, we first established an orthotopic mammary cancer model by inoculating the left abdominal mammary fat pad of mice with 4T1 tumor cells. To measure both innate and adaptive immune responses, we treated the tumor-bearing mice with Ausome twice at 7-day interval. To induce local hyperthermia, 30 min of laser irradiation was applied to the tumor region 6 h post injection. Injection of AuNP and subsequent laser irradiation served as a hyperthermia-only control. After the second treatment, we evaluated the levels of immune-related cytokines and chemokines in tumor tissue. Indeed, the various treatments elicited differential expression profiles of these proteins, as exhibited in the heat map in Fig. 5b. Ausome without laser irradiation (AS) triggered a marked increase in immune activation-related cytokines and chemokines, including TNF-a, IL-2, IL-6 IL-1a, CXCL13, and slightly upregulated IL-22, IL-1b, IL-17, IL-23, IL-27 as well as the T cell and NK cell recruitment-related chemokines, CCL4, CCL20, CCL3, CCL2 (Fig. 5b, Supplementary Fig. 17a–d). Ausome with laser irradiation (ASL) further augmented the intratumoral contents of these cytokines and chemokines (Fig. 5b, Supplementary Fig. 17a–d), indicating that the added heating potentiated the immune responses. In comparison, hyperthermia alone with AuNP (ANL) elicited only a weak modulation of cytokine levels (Fig. 5b). Other than the super-position of heat-induced immune effects, hyperthermia enhanced perfusion of excess Ausome may likewise contribute to the ASL-reinforced immunity (Fig. 5c). Significantly elevated cytokine levels

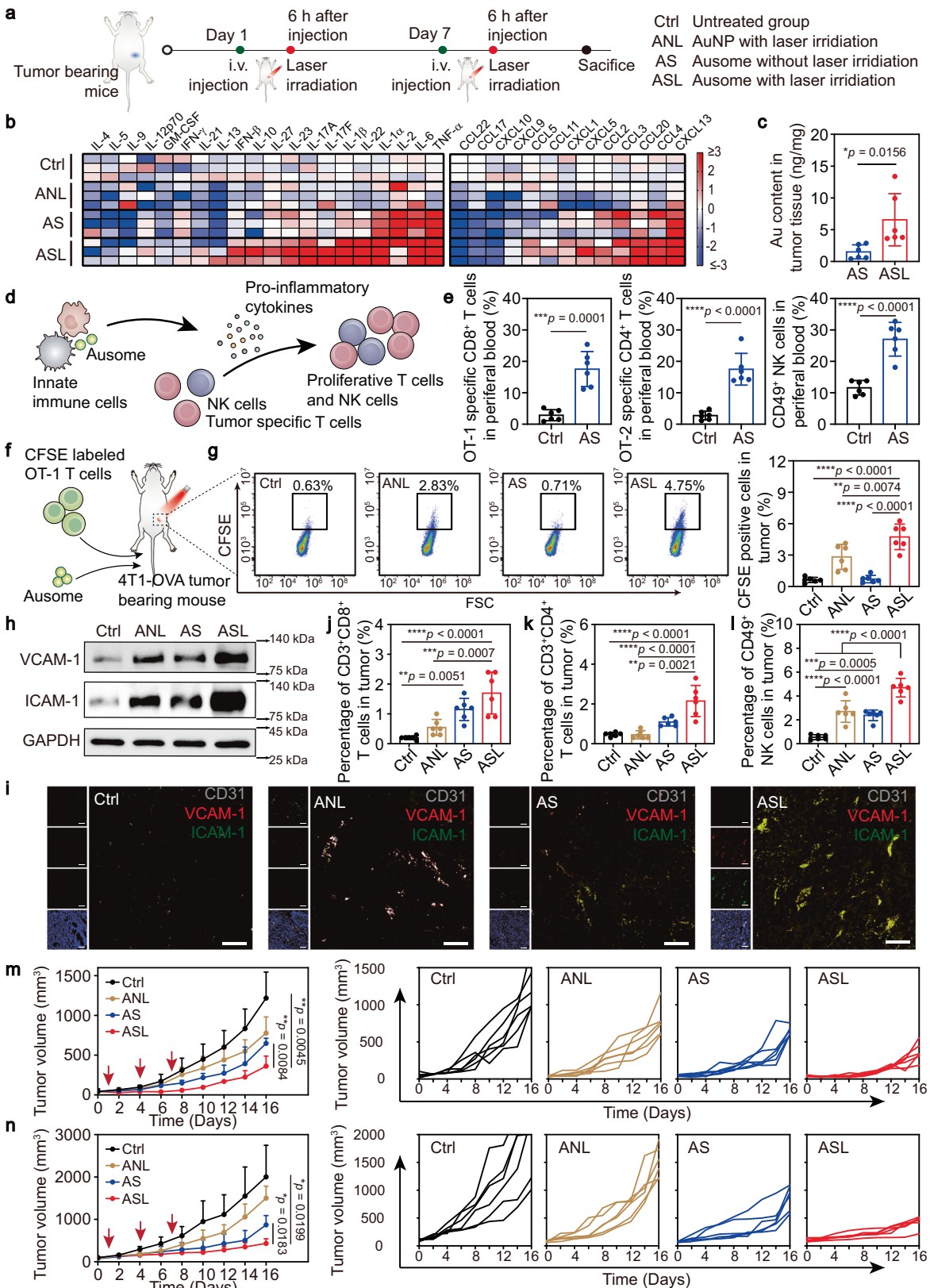

were also detected in the serum of the AS and ASL groups (Supplementary Fig. 18a–e), indicating the mobilization of systemic immune responses by Ausome.

The type, density, and distribution of immune cells within tumors have been closely associated with cancer patient prognosis, with infiltration by an abundance of effector lymphocytes usually indicative

of long-term survival and improved response to treatment[6]. Therefore, we analyzed the infiltration of effector immune cells to evaluate the ASL strategy elicited antitumor immunity. To better demonstrate the immune cell responses occurring at different stages in ASL-regulated immunity, we measured Ausome-triggered expansion of lymphocytes and local hyperthermia-promoted intratumoral infiltration

**Fig. 5 | Ausome-mediated multi-layered modulation of intratumoral immune contexture and tumor inhibition. a** Schematic illustration of the treatment procedures and the labels used for each regimen. **b** Heat map of the relative levels of the indicated cytokines in tumor tissues (n = 3 mice). Changes of 3-fold or greater times are displayed with the same maximal intensity. **c** ICP-MS quantified Au content in Ausome-treated tumors before and after laser exposure (n = 6 mice). **d** Schematic illustration of the procedure of Ausome activating effector lymphocytes. **e**, Flow cytometry analysis of OT-1 specific CD8+ T cells, OT-2 specific CD4+ T cells and NK cells in peripheral blood after treating with Ausome (n = 6 mice). **f**, Experimental design of local hyperthermia promoted effector lymphocyte infiltration in tumor tissue. **g**, Representative flow cytometry dot plots and statistical chart of CFSE positive cells in tumors (n = 6 mice). **h**, Immunoblotting of total tumor protein with antibodies against VCAM-1 and ICAM-1. **i**, Representative confocal microscopy images of tumor sections that were immunofluorescence-stained with antibodies against CD31, VCAM-1 and ICAM-1 (blue: nucleus, grey: CD31, red: VCAM-1, green: ICAM-1). Scale bars, 100 μm. **j–l**, Flow cytometry analysis of CD3+CD8+ T cells (**j**), CD3+CD4+ T cells (**k**) and CD49+ NK cells (**l**) in tumors isolated from mice (n = 6 mice) treated with PBS, ANL, AS or ASL, at 24 h postlaser irradiation. The average and individual tumor volumes of the 4T1 breast cancer (**m**) and CT26 colon cancer (**n**) models during the therapeutic experiment (statistical comparison performed on day 16; n = 6 mice) are shown. Red arrows indicate the times of treatment. The numerical data in (**c**, **e**, **g–j**) and (**m**, **n**) are shown as the mean ± s.d. *$p < 0.05$, **$p < 0.01$, ***$p < 0.001$, ****$p < 0.0001$; Two-tailed unpaired t-test (**c**, **e**), one-way analysis of variance (ANOVA) (**g–j**) or two-way ANOVA (**m**, **n**) followed by the Bonferroni multiple comparison test were used for statistical significance analysis. These experiments (**h**, **i**) were repeated three times independently with similar results. Source data are provided as a Source Data file.

respectively. As schemed in Fig. 5d, after intravenous injection, Ausome could rapidly be recognized by innate immune cells, which then mature to release pro-inflammatory cytokines (Supplementary Fig. 8b, Fig. 11d, e, Fig. 18a–e) and promoted effector immune cells activation and proliferation. To evaluate tumor-specific T-cell responses, we used a 4T1 breast cancer cell expressing ovalbumin (4T1-OVA) to establish the mouse tumor model. The levels of specific CD8+ and CD4+ T cells for MHC-I epitope OT-1 and MHC-II epitope OT-2 of OVA in peripheral blood were measured then. Compared to the tumor-bearing mice in the non-treated control group, Ausome treatment markedly increased the tumor-specific T cells, with over 6 times more OT-1 specific CD8+ T cells detected (Fig. 5e). Significant elevation of another important immune population, NK cells, was also observed within Ausome treatment (Fig. 5e), indicating a comprehensive proliferation or expansion of effector immune cells. The promoted tumor infiltration of these circulating effector lymphocytes by Ausome-mediated local hyperthermia was investigated then. As presented in Fig. 5f, 4T1-OVA bearing mice were immunized with Ausome twice as described above. 2 h before the second irradiation, carboxyfluorescein succinimidyl ester (CFSE) labeled OT-1 T cells, which was isolated from a transgenic mouse with TCR populations that specifically recognized OT-1 bound to H-2Kb complex, were intravenously injected. The tumor infiltrated CFSE-positive cells were measured using flow cytometry. We observed improved OT-1 cell accumulation in tumor tissue with local hyperthermia treatment. At the same time, Ausome-mediated thermal therapy further facilitated this phenomenon (Fig. 5g), which suggested other mechanisms in hyperthermia promoted immune cell infiltration besides blood perfusion. To gain additional insights into the mechanisms stimulating immune cell infiltration after ASL treatment, we investigated the expression of cell adhesion molecules that are often aberrantly downregulated on the tumor vasculature, consequently reducing the adhesion of immune cells and tumor penetration[44]. Western blot analysis revealed that both ANL and AS treatments elicited higher levels of vascular cell adhesion molecule-1 (VCAM-1) and intracellular adhesion molecule-1 (ICAM-1) in tumor tissue, especially on the surface of CD31+ tumor vascular cells (Fig. 5h, i). In comparison, ASL further upregulated the expression of ICAM-1 and VCAM-1 (Fig. 5h, i), which is expected to promote the retention of lymphocytes, and amplify antitumor immunity. We then evaluated the infiltrated effector immune cells, including T cells and NK cells, via flow cytometry analysis and immunostaining. No significant increases in T cells were observed in the ANL group (Fig. 5j, k), while AS elicited an apparent proliferation of T cells, with an approximately 5.75 times greater proportion of CD3+CD8+ T cells in tumor tissues, when compared to the untreated group (Fig. 5j, Supplementary Fig. 19, 20a). Additional local hyperthermia further facilitated T cell infiltration, achieving an approximately 9-fold increase in the levels of CD3+CD8+ T cells and 5-fold more CD3+CD4+ T cells, compared to the untreated group (Fig. 5j, k, Supplementary Fig. 19, 20a, b). Similarly, an elevated number of NK cells was also observed in the AS-treated tumors, which

was further enhanced by combination with laser irradiation. ANL treatment also facilitated increased infiltration by NK cells, but these cells were not of the activated, CD69+ phenotype (Fig. 5l, Supplementary Fig. 20c, d). Increased infiltration of macrophages was also detected after ANL or ASL treatment (Supplementary Fig. 20e). The intratumoral levels of other immune cells, such as B cells, dendritic cells, immune suppressive Tregs and myeloid-derived suppressor cells (MDSCs), showed no noteworthy alterations (Supplementary Fig. 20f–i).

By unleashing pro-inflammatory cytokines and enhancing effector lymphocyte infiltration, the ASL strategy led to an improved immune contexture in the tumor. We next evaluated whether these changes indeed resulted in tumor inhibitory effects. To accomplish this, we generated tumor-bearing mice by in situ inoculation with 4T1 breast cancer cells. After the tumors reached approximately 100 mm³, we intravenously injected the animals with Ausome at 15 mg/kg, 3 times at a 3-day interval. For the ASL strategy, 6 h after each injection, the tumors were treated with local hyperthermia via 30 min laser irradiation. The tumor sizes were measured every other day. To assess the therapeutic effects of hyperthermia alone, we applied AuNP following the same procedure described above. This strategy resulted in slightly controlled tumor growth, but without a significant difference when compared to the untreated group (Fig. 5m), while the laser alone elicited little suppression on tumor growth (Supplementary Fig. 21), indicating the slight tumor inhibition was the local hyperthermia resulted effects. In comparison, Ausome alone elicited obvious suppression on the established tumors, and additional hyperthermia further amplified this antitumor efficacy, substantially enhancing the observed tumor growth inhibition (Fig. 5m). We also used a subcutaneous CT26 colon tumor model to assess the antitumor effects of the Ausome-mediated multi-layered modulation. Compared to the mild disruption of AS treatment alone on the rapidly enlarged tumor, the combination of AS and hyperthermia led to markedly reduced tumor growth (Fig. 5n), which validates the therapeutic potential of the ASL strategy. In conclusion, the intrinsic photothermal conversion and immunostimulation abilities of Ausome stimulated an improved immune contexture in the tumor, including elevated levels of pro-inflammatory cytokines and effector immune cells, which led to enhanced tumor inhibition. This multi-layered modulation strategy significantly amplified the Ausome-induced antitumor immunity, with potential mechanisms as follows: 1) hyperthermia-mediated blood perfusion facilitates Ausome accumulation in tumor tissue, which is associated with increased cytokine and chemokine release, thus enhancing the activation, proliferation and recruitment of effector lymphocytes, 2) excess blood perfusion under hyperthermia buttresses immune cell accumulation and permeation into the tumor region alongside the bloodstream and 3) additional hyperthermia further increases the expression of cell adhesion molecules on vascular endothelial cells,

contributing to better adhesion and retention of immune cells at the tumor site.

## Ausome-mediated multi-layered modulation enhances the sensitivity of tumors to well-established immunotherapy and chemotherapy

Our increasingly robust knowledge of the interactions between tumors and the immune system has recently enabled the effective stratification of patients, as well as insights into the clinical limitations of many of the existing therapeutic strategies, including immune checkpoint blockade (ICB) and immune response-involved chemotherapies[6]. Therefore, the simultaneous application of immunomodulatory strategies aimed at manipulating the tumor microenvironment to a more pro-immune state may be a viable approach to enhance the final antitumor potency. To investigate whether Ausome can mediate an auxiliary enhancement of antitumor immunotherapy, we administered a combination of blocking antibodies against programmed cell death-1 (PD-1) and ASL. As shown in the schematic illustration (Fig. 6a), we intravenously injected 4T1 breast cancer-bearing mice with Ausome and treated the animals with the multi-layered modulating procedure described above; the anti-PD-1 antibody was intraperitoneally administrated every other day. The blocking antibodies alone elicited a limited inhibition of the overall tumor growth, likely due to the pre-existing barren microenvironment, which lacked sufficient effector T cell infiltration (Fig. 6a–d). In contrast, the combined antibody and ASL treatment significantly reduced tumor size, compared with the untreated group or anti-PD-1 antibody monotherapy group (Fig. 6b, c), suggesting that Ausome-mediated expansion and accumulation of lymphocytes in the tumor region may have provided a fertile ground for the immune utilization of the PD-1 blocking agent (Fig. 6d). We also observed an upregulation of PD-L1 in tumor tissue after treatment with ASL, presumably because of the heightened immune pressure elicited by ASL, which was then blocked by the anti-PD-L1 antibody (Fig. 6e).

Apart from the emerging immunotherapies that are highly dependent on the host's own immune system, some traditional chemotherapeutic agents, such as doxorubicin (Dox), have also recently been reported to simultaneously elicit T cell responses while killing tumor cells, and undergo antitumor efficiency enhancement by immunomodulation[45,46]. Thus, we applied Ausome to improve the tumor baseline immunity and possibly promote Dox-induced immunity, to enhance overall antitumor potency. Again, we followed the Ausome-mediated multi-layered modulation procedure described above, with 1.5 mg/kg Dox intravenously injected before each laser irradiation treatment (Fig. 6f). Compared with the control group, free Dox elicited an increasing release of intratumoral, high-mobility group box-1 (HMGB1), as well as overexpression and exposure of calreticulin (CRT) on the surface of tumor cells (Fig. 6g). These two molecules are immunogenic eat me and danger signals that play vital roles in triggering T cell responses. In contrast, in the ASL-modulated tumors, Dox triggered a more notably elevated fluorescence intensity of CRT and HMGB1 release (Fig. 6g), likely due to both the adjuvant properties of Ausome and a heat-facilitated increase in tumor Dox perfusion (Fig. 6g, Supplementary Fig. 22). After the Ausome-elicited, multi-layered immune modulation and Dox administration, we isolated the tumors and assessed the degree of T cell infiltration by immunofluorescence staining and flow cytometry analysis. At this low dose, only a limited number of CD4+ and CD8+ T cells were detected in free Dox-treated tumors (Fig. 6h, i). By contrast, combined ASL and Dox treatment markedly increased the infiltration of T cells, compared with the free Dox group (Fig. 6h, i). In the combined therapy-treated tumors, we also detected a greater percentage of mature DCs (Supplementary Fig. 23), which expressed higher levels of CD80 and CD86. These co-stimulatory molecules are responsible for effective tumor antigen presentation and thus contribute to the enhanced degree of intratumoral T cells. Having verified the increased accumulation of T cells in tumors with the combined chemo-immune treatment, we proceeded to evaluate the therapeutic potential of the strategy. As shown in the treatment scheme in Fig. 6f, we treated breast tumor-bearing mice for three cycles with Dox and/or ASL procedures. Dox monotherapy elicited only a modest tumor suppression and no T cell response, both of which were consistent with the relatively limited dosage (Fig. 6j, k). In contrast, a robust tumor regression was apparent in the groups treated with ASL plus Dox (Fig. 6j, k). In light of the severe side effects often triggered by Dox chemotherapy, such as irreversible heart injury, we examined the effects on the heart of our low-dose and low-frequency regimen that effectively treats tumor-bearing mice. After three treatments, we assessed the morphology of the heart by H&E staining of tissue sections. Notably, no damage was observed in both the free Dox-treated mice and the chemo-immune combined therapy-treated mice (Supplementary Fig. 24), indicating that good therapeutic outcomes can be achieved by the low dose of Dox with ASL, eliminating the safety concerns associated with high doses of chemotherapeutic agents. In conclusion, our findings indicate that Ausome-mediated immune modulation can significantly enhance the sensitivity of tumors to ICB antibodies, as well as chemotherapeutic agents, which may extend the clinical application of these well-established agents.

## Discussion

In the present study, we describe a biologically generated nanomaterial that manipulates the tumor immune microenvironment via multi-layered modulation to reinforce antitumor immunity, and potentiates the sensitivity of tumors to several types of clinical, first-line therapies. Recent studies have demonstrated an important involvement of the immune system in tumorigenesis and progression. Based on this, multitudes of agents have been screened and tested for the ability to elicit potent antitumor effects through resuscitating the weakened immunity in the tumor region or alleviating immune suppression[3,4]. To address the deficiency or dysfunction of intratumoral effector lymphocytes, varies strategies, such as immune agonists, chemokines or cytokines (e.g., TLRs ligands, TNF-a, CCL2, IL-2) have been developed and assessed for their abilities in facilitating antigen specific responses, recruitment of effector immune cells, or sustaining the activity of these cells, in clinical and/or pre-clinical trials[6,11]. However, modulating only a single signaling pathway is likely to be compromised by other immune suppressive mechanisms or induced tolerance, as well as introduces the possibility of systemic toxicity[17]. Artificial nanomaterials can be exploited to integrate immunomodulating molecules and deliver therapeutics in vivo, however, great challenges remain to more precisely control the types and amounts of the loaded cargoes[14]. Natural materials, such as bacteria-derived outer membrane vesicles (OMVs), are aggregations of diverse immune agonists (e.g., LPS, lipoproteins and other immunogenic proteins) and can stimulate potent immune responses[14,16]. In addition, these materials are directly generated from bacterial cells, which, in contrast to fully artificial nanomaterials, avoids stringent chemical reactions and arduous multi-component assembly steps, therefore enabling facile bulk fabrication. These distinct advantages make bacteria-derived materials excellent candidates for tumor immune state remodeling.

The natural immunomodulator, Ausome, we describe in this study was generated in vivo by *E. coli*, and comprises a bacterial component shell surrounding a gold nanoparticle inner core. These components endow Ausome with multi-functional features and potentials to stimulate comprehensive and selective immune reactions, as shown in Fig. 1. The diverse bacterial molecules on Ausome initiated a systemic immune response, as well as a response local to the tumor, and triggered the secretion of a broad spectrum of cytokines and chemokines that stimulated pro-inflammatory immune cell activation, expansion and migration. These immune responses were then specifically amplified in the tumor region under local hyperthermia, mediated by

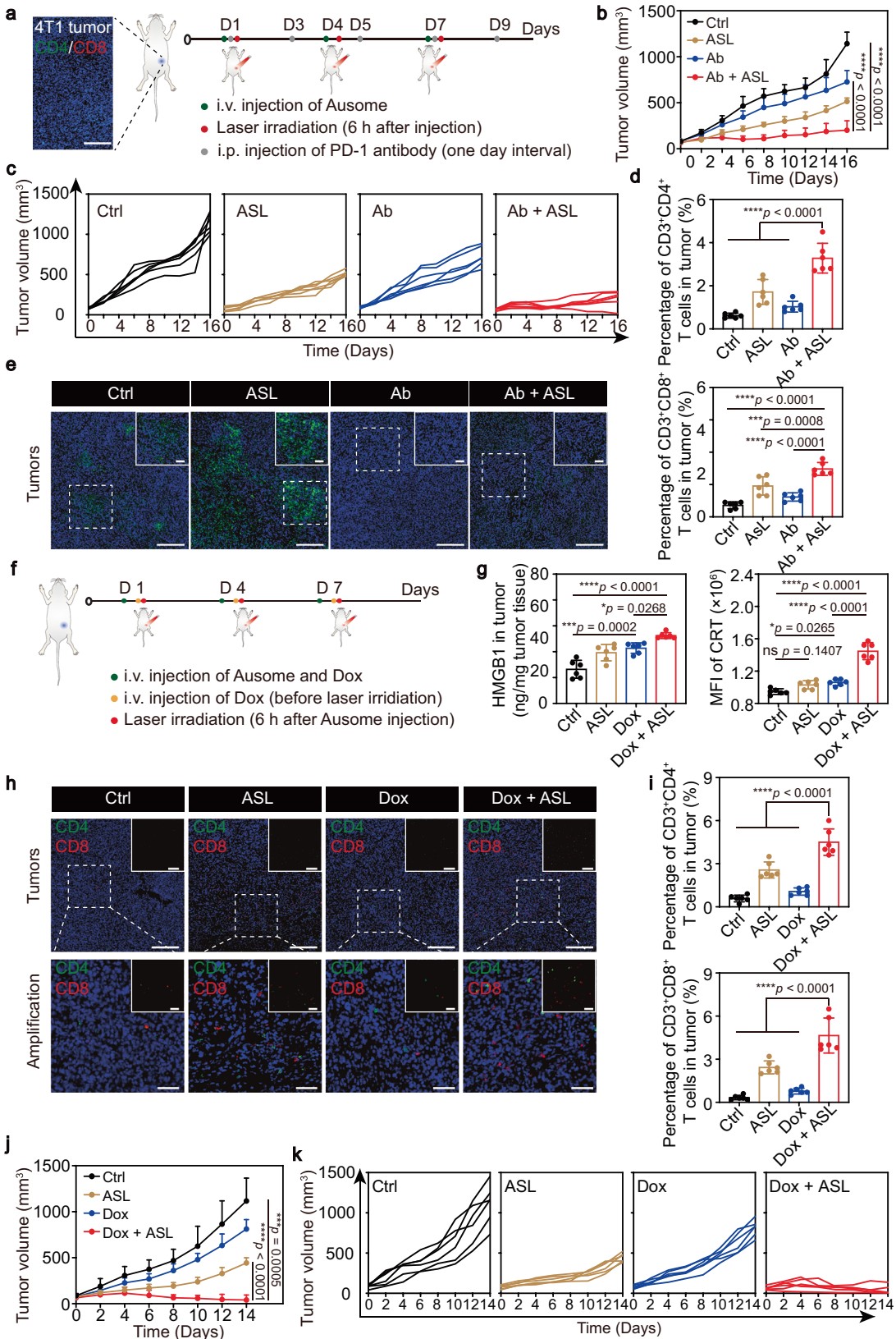

the gold nanoparticles within Ausome, through the facilitated infiltration of both Ausome nanoparticles and effector immune cells along with enhanced blood flow into the tumor tissue. Such multi-layered modulation consequently, selectively remodeled the immune contexture of the tumor region, which led to enhanced antitumor therapeutic performance and alleviated the safety concerns of systemic administration of a large amount of danger signals. In addition to affecting local hyperthermia, the gold nanoparticle inner core also afforded additional mass to the nanoparticle, which enabled to the concentration of Ausome by high-speed centrifugation, without necessitating an ultracentrifuge, which is common in the isolation of other bacteria-derived nanomaterials, such as OMVs.

**Fig. 6 | Ausome-mediated improvement of the tumor immune microenvironment, and enhanced sensitivity to immunotherapy and chemotherapy.**
**a** Representative confocal microscopy image of a 4T1 tumor immunofluorescence-stained with antibodies against CD4 (red) and CD8 (green), and scheme of the therapeutic experimental design to assess antitumor efficiency of PD-1 blockade (Ab) with ASL-induced immune regulation. Scale bar, 200 μm. **b, c** The average (**b**) and individual (**c**) tumor growth curves in mice (*n* = 6 mice) after treatment with ASL, Ab monotherapy or combined ASL and Ab treatment (ASL + Ab). **d** Flow cytometry analysis of the percentage of CD3$^+$CD4$^+$ and CD3$^+$CD8$^+$ cells in tumors (Day 16; *n* = 6 mice). **e** Representative confocal microscopy images of the PD-L1 levels in tumor tissues (blue: nucleus, green: PD-L1; day 16). Inset: enlarged images. Dashed white boxes indicate the enlarged zone. Scale bars, 200 μm and 50 μm (enlarged images). **f** Schematic illustration of the therapeutic experimental procedure to assess the antitumor efficacy of doxorubicin (Dox) in ASL-modulated tumors. **g** The amount of released HMGB1 per tumor mass and CRT levels in tumor tissues (day 2; *n* = 6 mice) as measured by ELISA or flow cytometry.
**h, i** Representative confocal microscopy images of tumor sections stained with antibodies against CD4 and CD8 (blue: nucleus, green: CD4, red: CD8, **h**), and flow cytometry-quantified CD3$^+$CD4$^+$ cells and CD3$^+$CD8$^+$ cells in tumors (**i**, day 14; *n* = 6 mice). Inset: confocal microscopy images without nuclear fluorescence. Dashed white boxes indicate the enlarged zone. Scale bars, 200 μm (upper panel) and 50 μm (lower panel). **j, k** Average (**j**) and individual (**k**) tumor volumes over the therapeutic experiment (statistical comparison performed on day 14; *n* = 6 mice). The numerical data in (**b–d, g, i–k**) are shown as the mean ± s.d. *$p < 0.05$, **$p < 0.01$, ***$p < 0.001$, ****$p < 0.0001$; one-way ANOVA (d, g, i) or two-way ANOVA (b, j) followed by the Bonferroni multiple comparison test were used for statistical significance analysis. These experiments (**e, h**) were repeated three times independently with similar results. Source data are provided as a Source Data file.

Ultracentrifugation is difficult to implement in industrial production. Importantly, the preparation of Ausome was through eco- and industry-friendly methods.

Our increased comprehension of the sustained interactions between tumor tissue and the immune system not only contributes to deeper insights on tumor progression, but also helps to reveal the mechanisms responsible for the erratic treatment outcomes of several clinical therapies, thus divulging strategies to optimize the antitumor potency of these well-established agents[6,47]. The ICB drugs, such as antibodies against PD-1/PD-L1, which were designed to unleash the power of T cells against malignant cells, have exhibited powerful tumor elimination ability in a subset of patients[12]. The failure of ICB in a significant cohort of individuals is largely attributed to insufficient intratumoral infiltration of effector immune cells, suggesting that the therapeutic potency of ICB relies heavily on some level of pre-existing immunity[6]. We therefore assessed the antitumor efficiency of a PD-1 blocking antibody in Ausome-modulated tumors, achieving considerable improvement in tumor inhibition compared to antibody monotherapy. Some traditional chemotherapeutic agents, such as doxorubicin, are widely applied in various clinical scenarios for extended treatment periods but are also well-known for their dose-dependent and severe side effects[46]. Many of these agents have then been reported to elicit T cell responses while killing tumor cells to exhibit enhanced therapeutic efficiency when provided in combination with immunomodulators[45]. We utilized Ausome to manipulate the tumor immune contexture into an immune-promoting state, which promoted antigen-presenting cell activation, a magnified T cell response and excellent tumor suppression, even when Dox was applied at a low dose. These findings offer alternative strategies to improve the therapeutic effect of these agents already in use.

Mineralized structures are formed by organisms through the process of biomineralization, which dates back about 570 million years ago[48]. Minerals synthesis by microbiota is an environmentally friendly reaction system, without the participation of toxicity chemicals, and can be regarded as green chemistry, showing great potential in biological, biochemical, and biomedical applications as well as industrial benefits. However, bacteria-driven biomineralization is most reported in controlling metal fixation, transport, speciation, and even toxicity in ecological environments[49,50]. In most cases, researchers are only focus on the functions of minerals themselves, since components of microorganisms (proteins, phospholipids, or nucleic acids) attached to the minerals are treated as impurities and hard to remove[51]. As a result, the application of the minerals obtained from microbiota is greatly restricted. In this study, we present a comprehensive investigation of the therapeutic potential of biomineralization-generated nanoparticles, both the mineral portion and the attached biological molecules, which broadens the functions and expands the applications of microorganism-mineralized products.

In summary, we demonstrate a bacterium-generated immunomodulator that selectively remodels the tumor environment via multi-layered regulation, thus improving the immune surveillance of malignant cells and enhancing the sensitivity of tumors to several well-established therapies, either by elevating baseline immunity or amplifying drug-induced immune responses. Other than Ausome mediated mono- or combinational therapies we described in this study, there are still a lot of room to fully exploit its application fields and therapeutic potentials. Through genetic engineering or chemical conjugation, Ausome could be modified with specific targeting ligands, to promote its accumulation in tumor tissue, thus enhancing the ultimate anti-tumor therapeutic efficacy and optimizing the laser irradiation conditions, such as lower laser power density. Gao et al. recently reported a strategy of delivering *E. coli* OMV-coated gold nanoparticles to inflammatory tumor tissues by in vivo hitchhiking phagocytic immune cells, with ingenious design of intracellular self-assembly to minimize exocytosis of nanoparticles from immune cells[30]. Gao's study exhibits another alternative for tumor targeted transport of Ausome, of which the clever design of gold nanoparticles aggregation could also lead to a red-shift of the maximum absorption, therefore, a longer wavelength laser with increased penetration depth could be used to inspire the photothermal effect of Ausome. In addition, the bacterium participated generation process, diverse immune stimulating components and gold inner core of Ausome make it an editable, immune system activable, high-Z material, which could be further utilized to design desirable anti-tumor strategies, including but not limited to fabricate potent cancer vaccine, custom-make delivery platform for protein or nuclei acid cargoes, or develop multifunctional radiosensitizer with intrinsic immune initiation features, all of this imply wide application of Ausome in diverse clinical scenario. Moreover, the size, morphology, surface modification, composition and yield can all be optimized in the future by screening optimal strains and/or editing the bacteria to satisfy different application demands and further drive the clinical translocation of this promising nanomaterial.

## Methods
Our research complies with all relevant ethical regulations. All animal experiments were approved by the Institutional Animal Care and Use Committee of the National Center for Nanoscience and Technology.

### Materials
Chloroauric acid (HAuCl$_4$) was purchased from Hushi (Shanghai, China; catalogue no: 10010711). Oxoid™ Yeast Extract Powder (catalogue no: LP0021B) and Oxoid™ Tryptone (catalogue no: LP0042B) were purchased from Thermo Fisher Science (Massachusetts, USA). RPMI 1640 medium, fetal bovine serum (FBS), penicillin/streptomycin, phosphate buffered saline (PBS) and ACK Buffer were obtained from Wisent (St. Bruno, Canada). L-Glutamine (catalogue no: 25030), β-mercaptoethanol (β-ME, catalogue no: 21985032) and sodium pyruvate (catalogue no: 11360070) were purchased from Gibco (Grand Island, USA). Granulocyte macrophage colony stimulating factor (GM-

CSF; catalogue no: CYT-282-2) and interleukin-4 (IL-4; catalogue no: CYT-222-5) were purchased from ProSpec (Ness-Ziona, Israel). IL-2 (catalogue no: 212-12-100UG) was obtained from PeproTech (Cranbury, USA). The peptides, OVA$_{257-264}$ (SIINFEKL) and OVA$_{323-339}$ (ISQAVHAAHAEINEAGR) were purchased from ChinaPeptides Co., Ltd (Shanghai, Chian). The golgi inhibitor GolgiPlug (catalogue no: 555028) was purchased from BD Bioscience (New Jersey, USA). The primary antibodies, the FITC-anti-mouse CD11c (catalogue No: 117305, clone name: N418, 1:100), PE-Cy7-anti-mouse CD80 (catalogue No: 104734, clone name: 16-10A1, 1:100), APC-anti-mouse CD86 (catalogue No: 105116, clone name: PO3, 1:100), FITC-anti-mouse CD3 (catalogue No: 100203, clone name: 17A2, 1:100), PE-anti-mouse CD4 (catalogue No: 100407, clone name: GK1.5, 1:100), PE-Cy7-anti-mouse CD8 (catalogue No: 100722, clone name: 53-6.7, 1:100), PE-anti-mouse-CD69 (catalogue No: 104508, clone name: H1.2F3, 1:100), Brilliant Violet 605-anti-mouse-CD25 (catalogue No: 102035, clone name: PC61, 1:100), APC-anti-mouse CD19 (catalogue No: 152410, clone name: 1D3/CD19, 1:100), FITC-anti-mouse CD49 (catalogue No: 108905, clone name: DX5, 1:100), PE-Cy7-anti-mouse-CD69 (catalogue No: 104512, clone name: H1.2F3, 1:100), APC-anti-mouse-CD11c (catalogue No: 117310, clone name: N418, 1:100), FITC-anti-mouse CD4 (catalogue No: 100405, clone name: GK1.5, 1:100), PE-Cy7-anti-mouse CD25 (catalogue No: 101916, clone name: 3C7, 1:100), Alexa Fluor 647-anti-mouse Foxp3 (catalogue No: 126408, clone name: MF-14, 1:100), FITC-anti-mouse F4/80 (catalogue No: 123108, clone name: BM8, 1:100), PE-anti-mouse-CD11b (catalogue No: 101208, clone name: M1-70, 1:100), Alexa Fluor 594-anti-mouse Gr1 (catalogue No: 108448, clone name: RB6-8C5, 1:100) were purchased from Biolegend (San Diego, USA). The anti-mouse Hsp 70 (catalogue No: ab2787, clone name: 5A5, 1:100), anti-mouse Hsp 90 (catalogue No: ab32568, clone name: E296, 1:100), anti-mouse ICAM-1 (catalogue No: ab222736, clone name: EPR22161-284, 1:1000), anti-mouse VACM-1 (catalogue No: ab134047, clone name: EPR5047, 1:5000) were purchased from Abcam (Cambridge, UK). Anti-mouse CD49 (catalogue No: 142602, clone name: HMα1, 1:100), anti-mouse CD31 (catalogue No: 102402, clone name: 390, 1:100) were purchased from Biolegend (San Diego, USA). Anti-mouse PD-L1 (catalogue No: 28076-1-AP, 1:500) were purchased from Proteintech (Manchester, UK). Alexa Fluor 488-anti-mouse calreticulin (catalogue No: 142602, 1:100) was obtained from Cell Signaling Technology (CST, Massachusetts, USA). The InVivoPlus anti-mouse PD-1 blocking antibody (catalogue No: BP0146, clone name: RMP1–14, 7.5 mg/kg for in vivo use) was purchased from BioXCell (Lebanon, USA). The cytokine quantification panels, LEGENDPlex T helper cytokine multiplex assay (catalogue no: 740005), LEGENDPlex inflammation cytokine multiplex assay (catalogue no: 740446), LEGENDPlex proinflammatory chemokine multiplex assay (catalogue no: 740451) were purchased from Biolegend (San Diego, USA). The mouse HMGB1 enzyme-linked immunosorbent assay (ELISA) Kit (catalogue no: SEKM-0145) was purchased from Solarbio (Beijing, China). The Ficoll Pague Plus solution (catalogue no: 17-1440-03) was purchased from GE (Boston, USA). The lymphocyte separation medium (7211011) was obtained from Dayou (Jiangxi, China). The carboxyfluorescein succinimidyl ester (CFSE, catalogue no: C34570) was purchased from Invitrogen (California, USA). All other chemicals used in this study were obtained from Sigma-Aldrich (St. louis, USA) unless otherwise noted.

## Mice and cells

6–8 weeks-old female Balb/c mice, 6–8 weeks-old female and male Sprague Dawley rats, were purchased from Vital River Experimental Animal Technology Co. Ltd (Beijing, China), 6–8 weeks-old female Jackson Laboratories sourced transgenic mice [C57BL/6-Tg (TcraTcrb) 1100Mjb/J] with MHCI restricted and OVA$_{257-264}$ specific TCR populations were purchased from Aniphe Biolaboratory Inc. (Jiangsu, China). Sex was not considered in this study, because this study aimed to demonstrate the photothermal effects, immunoregulating abilities and anti-tumor outcomes of Ausome, which should not be significantly affected by sex. We used female animals in most experiments was for generating orthotopic breast cancer model, which is a superficial tumor and suitable for verify Ausome mediated laser dependent photothermal effects. Mice were raised under pathogen-free conditions with temperature of 20–22 °C and humidity of 30–70%. The artificial light was set in a 12 h light and 12 h dark cycle. Food and water were always available for the animals. All animal-involving procedures were performed in compliance with experimental guidelines approved by the Animal Care and Welfare Committee of the National Centre for Nanoscience and Technology. The maximal tumor size permitted by the ethics committee is no more than 20 mm (the longer axis of tumor), and none of the tumors in this study exceeded this size limit. The 4T1 mouse mammary cancer cells (catalogue no: CRL-2539), mouse colon carcinoma CT26 cells (catalogue no: CRL-2638) and human umbilical vein endothelial cells (HUVEC, catalogue no: PCS-100-010) were purchased from the American Type Culture Collection (ATCC, MA, USA) and cultured in RPMI 1640 medium supplemented with 10% FBS and 100 μg/mL penicillin/streptomycin and maintained in 5% CO$_2$ at 37 °C. The 4T1 mouse breast cancer cells expressing ovalbumin (OVA, 4TI-OVA, catalogue no: CE27569) was obtained from Keruisibo (Beijing, China). Bone marrow-derived dendritic cells (BMDCs) were generated from murine bones as previously described. Briefly, bone marrow was harvested from isolated femurs and tibias via flushing with medium after removal of both distal ends of the bones. The samples were then incubated with ACK lysis buffer for 1 min to lyse red blood cells and centrifuged at 300× $g$ for 5 min to collect the remaining cells. The collected cells were dispersed into RPMI 1640 medium supplemented with 10% FBS, 100 μg/mL penicillin/streptomycin, 2 mM L-glutamine, 0.05 mM β-ME, 20 ng/mL GM-CSF and 10 ng/mL IL-4 and cultured in 6-well plates at a density of $2 \times 10^6$ cells per well. Fresh medium was provided every 2 days. After 6 days of culture, the non-adherent cells were collected for further investigation. To induce bone marrow-derived macrophages (BMDMs), the bone marrow was prepared as described in BMDCs preparing, which were then cultured in RPMI 1640 medium supplemented with 10% FBS, 100 μg/mL penicillin/streptomycin, 2 mM L-glutamine, 0.05 mM β-ME, 5% 1-week L929 cell conditioned medium and 5% 2-week L929 cell conditioned medium and cultured in 6-well plates at a density of $2 \times 10^6$ cells per well. Fresh medium was supplied every 2 days and cells were cultured for 6 days before use.

## Ausome preparation

100 μL frozen *Escherichia coli* (*E. coli*) DH5α strain (Invitrogen, catalogue no: 18265017) was seeded into 200 mL lysogeny broth (LB) medium, followed by shaking at 200 rpm at 37 °C. After about 12 h growth, when the optical density at 600 nm (OD600) of the culture solution was approximately 1.0, the bacterial cells were isolated by centrifuging at 5000× $g$ for 10 min (Multifuge X1R, Thermo Fisher, USA), 4 °C. The supernatant was discarded and the precipitate was resuspended to single cells by vortexing in 20 mL PBS, followed by addition of 8 mL 1% (wt/vol) HAuCl$_4$ solution to the cell suspension. The total solution volume was adjusted to 200 mL with PBS, and the sample was then returned to the incubator to react at 37 °C with 200 rpm shaking. After incubating the samples for predetermined intervals, the secreted Ausome was collected through two steps of centrifugation. The bacterial cells were first removed by centrifuging the culture solution at 5000× $g$ for 10 min (Multifuge X1R, Thermo Fisher, USA) and the supernatant was sequentially filtered through polysulfone ether (PES) membranes with 0.45 μm and then 0.22 μm pore size. The germfree solution was subsequently centrifugated at 11,000× $g$ for 30 min (Multifuge X1R, Thermo Fisher, USA). After discarding the supernatant, the sediment was resuspended in PBS and washed once; the final product was dispersed in 1 mL PBS and stored at −80 °C for use. Proteomic analysis was performed to detect the protein composition of

Ausome, we mixed Ausome samples from three independent preparation process, and the mass spectrometry detection were entrusted to H·Wayen Biotechnology Co. Ltd, (Shanghai, China).

To explore the generation of Ausome in *E. coli*, bacterial cells were isolated before and after incubation with $HAuCl_4$ for 0.5, 4, 12, 24, 48 or 96 h, by centrifuging at $5000\times g$ for 10 min, followed by washing once in PBS and collection via another centrifugation step. Afterwards, the sediment was resuspended to single cells and dropped onto a carbon film-coated copper mesh or a hydrophilization-treated silicon wafer, and naturally dried at room temperature. The samples on copper mesh were examined using a transmission electron microscope at an acceleration voltage of 120 kV (TEM, Ht-7700, Hitachi, Japan), and the silicon wafer-loaded samples were sputtered with Au ions prior to detection using a scanning electron microscope (SEM, SU8220, Hitachi, Japan).

The cell pellet collected at 48 h was further sequentially fixed with 2.5% glutaraldehyde (overnight) and 1% osmium tetroxide (2 h), followed by dehydration in graded ethanol and embedding in epoxy resin. After polymerization at 45 °C and 60 °C, the hardened cell block was sliced to into ultrathin sections (70 nm) and loaded on a carbon film-coated copper mesh. After staining with uranyl acetate and lead citrate, the cell sections were observed under STEM mode (SU8220, Hitachi, Japan).

### Gold nanoparticle (AuNP) preparation (chemical synthesis)

30 nm AuNP was synthesized via the citrate reduction of chloroauric acid ($HAuCl_4$). Briefly, 1 mL 1% (wt/vol) $HAuCl_4$ solution was diluted into 100 mL distilled water and heated with magnetic stirring at 400 rpm in a 120°Coil bath for 5 min, followed by addition of 1 mL 10 mg/mL sodium citrate solution under heating and stirring; the reaction solution rapidly turned to dark red within 30 s. After reacting at 120 °C for 15 min, the solution was transferred to room temperature and stirred at 1100 rpm and then allowed to cool for 20 min. Next, 10 mg polyethylene glycol (2000 Da) modified with terminal sulfhydryl (PEG-SH) was added and the solution was stirred overnight. AuNP was collected by centrifuging at $11,000\times g$ for 30 min and washed once with distilled water. The final product was stored at 4 °C in the dark.

### Physicochemical characterization

The stock solution of Ausome was diluted in deionized water and dropped onto a carbon film-coated copper grid. The morphology of Ausome was examined by TEM at an acceleration voltage of 200 kV (Tecnai G2 F20 U-TWIN, FEI, USA), and the crystal structure of Ausome was imaged under high-resolution TEM mode. The selected area electron diffraction (SAED) pattern of Ausome was obtained using a TEM equipped with an energy dispersive spectroscope (EDS) attachment (Tecnai G2 F20 U-TWIN, FEI, USA). The hydrated size of Ausome (diameter, nm) was determined using a 633 nm He-Ne laser-equipped ZetaSizer Nano series Nano-ZS (Malvern, UK), and a Zetasizer Software (version 8.01.4906) to collect data.

Thermalgravimetric (TG) analysis was performed to detect the organic and inorganic contents within Ausome. Before lyophilization, the Ausome suspension was desalted using an ultrafiltration centrifuge tube with a molecular weight cut-off of 30 kDa; the weight loss of the Ausome powder was measured using by TGA (TG 290F3, Netzsch, Germany) at a heating rate of 10 °C/min from 34 °C to 600 °C.

### Photothermal performance

The photothermal effects of Ausome and AuNP were evaluated by measuring temperature changes under laser irradiation. Samples of different concentrations, including 500, 250, 125, 63, 31 and 15 μg/mL (defined by the Au content; quantified by inductively coupled plasma mass spectrometry [ICP-MS, Thermo-X7, Thermal, USA]), were exposed to a 660 nm continuous laser with a power density of 1.5 W/cm² for 10 min. The photothermal performance of Ausome and AuNP

was also tested by irradiation using different power densities, including 2.5, 1.5 1.0 and 0.5 W/cm², at a fixed concentration of 250 μg/mL. To compare the photothermal conversion efficacy, 250 μg/mL Ausome and 250 μg/mL AuNP solutions were irradiated using a 660 nm laser with a power density of 1.5 W/cm² for 10 min each. During the irradiation, an infrared thermal camera (FLUK-TiS60 + 9 Hz, Fluke, USA) was used to monitor the temperature changes in real time.

### Ausome-mediated immune effects in vitro

BMDCs ($1 \times 10^5$ per dish) were seeded into confocal microscopy dishes and cultured overnight, followed by addition of 10 μg Ausome or AuNP and incubation for 6 or 24 h. Afterwards, the BMDCs were washed twice with PBS and imaged using a two-photon laser confocal microscope (FV1000, Olympus, Japan).

To explore the interactions between Ausome and innate immune cells, as well as the potential mechanisms of Ausome-mediated immune responses, BMDCs ($1 \times 10^6$ per well) were seeded into 6-well plates and incubated with 100 μg Ausome for 24 h. The cells were then collected in RNAse-free tubes via centrifuging at $300\times g$ for 5 min and resuspended in 1 mL TRIzol. After snap-freezing in liquid nitrogen for 0.5 h, the samples were stored at -80 °C. Transcriptome sequencing was performed by Shanghai Majorbio Bio-pharm Technology Co., Ltd. (Shanghai, China), and the Ausome-triggered differentially expressed genes in BMDCs were identified by comparing the results with untreated BMDCs (mock group).

The Ausome-elicited activation and maturation of immune cells were tested by treating BMDCs with different doses of Ausome. Briefly, $2 \times 10^5$ BMDCs were dispersed into 1 mL RPMI 1640 medium supplied with 10% FBS and seeded into 1.5 mL sterile tubes and incubated with 10.0, 5.0, 2.5, 1.3, 0.6, 0.3 or 0 μg Ausome. 6 or 24 h later, the cells were collected and washed with FACS buffer (RPMI 1640 medium supplied with 2% FBS). Next, a FITC-labeled antibody against mouse CD11c, a PE-Cy7-labeled antibody against mouse CD80 and an APC-labeled antibody against mouse CD86 were added into the cell suspensions. After incubating for 20 min at 4 °C, the cells were collected and washed once in FACS buffer. The proportions of CD80 and CD86 on CD11c$^+$ cells were detected by flow cytometry (Accuri$^{TM}$ C6, BD, USA).

To investigate the activation of T cells by BMDCs induced by Ausome, $3 \times 10^5$ BMDCs were seeded into 1.5 mL sterile tubes and incubated with or without 1 μg Ausome. After 12-h incubation, BMDCs and culture supernatant were collected separately. The lymphocytes were isolated using lymphocyte separation solution, and co-cultured with harvested BMDCs at a ratio of 5:1 in a mixture of the obtained BMDCs culture supernatant and T cell medium (RPMI 1640 supplemented with 10% HI-FBS, 2 mM L-glutamine, 1 mM sodium pyruvate, 50 μM β-mercaptoethanol, 1% penicillin-streptomycin, 30 U/mL IL-2) for 24 h. Then CD69 or CD25 highly expressed T cells were detected using a flow cytometry (Accuri$^{TM}$ C6, BD, USA).

### Tumor accumulation and biodistribution of Ausome

6−8 weeks-old female Balb/c mice ($n = 3$ per group) were inoculated with 4T1 breast cancer cells into the left abdominal mammary fat pad. When the tumor size reached about 300 mm³, 15 mg/kg Ausome was intravenously injected into the mice. To track Ausome in vivo, the nanomaterial was pre-labeled with the fluorescent agent, Cy5.5. 0, 2, 6, 12, 24, 48 or 72 h postinjection, the distribution of Cy5.5 fluorescence in the mice was assessed using a Maestro in vivo spectrum imaging system (IVIS, CRi, Woburn, MA, USA). Fluorescence intensities at the tumor region were analyzed through Living Image 4.3 software.

To further quantify the tumor accumulation and biodistribution of Ausome, 4T1 tumor-bearing, 6−8 weeks-old female Balb/c mice ($n = 3$ per group) were established as described above, followed by intravenous injection of 15 mg/kg Ausome. In parallel, the same amount of AuNP was injected into tumor-bearing mice ($n = 3$). After 0, 2, 6, 12, 24, 48 or 72 h, the mice were euthanized and the tumors were

isolated and weighted. The major organs, including thymus, heart, lung, liver, kidney, spleen, lymph node and intestine, were also collected at 6 and 24 h. Prior to elemental quantification, the tissues were digested with $HNO_3$ and $H_2O_2$ at 150 °C, the resultant Au solutions were diluted with 2% $HNO_3$ and the samples were analyzed by ICP-MS (Thermo-X7, Thermal, USA).

### Evaluation of Ausome biosafety in rats
To evaluate the biosafety of Ausome, we then intravenously injected Ausome into Sprague-Dawley rats following the immune procedure performed in mice, that is 3 times administration at a 3-day interval. We chose a low dose of 6.5 mg/kg, which was conversed from dosage applied in mice and calculated according to the formula (1) and a higher dose of 13 mg/kg.

$$D_{Rat} = \frac{D_{Mouse} \times W_{mouse}}{K_{Mouse} \times W_{mouse}{}^{0.67}} \times \frac{K_{Rat} \times W_{Rat}{}^{0.67}}{W_{Rat}} \quad (1)$$

where D is dose of Ausome used in rats or mice (mg/kg), W is the body weight of rats or mice (g); K×W determined the surface area of animals, in which K is constant and varied in different species, $K_{mouse}$ is 9 and $K_{rat}$ is 9.6[52].

6–8 weeks old rats were divided into 3 groups ($n = 3$ female rats + 3 male rats), including 6.5 mg/kg Ausome treated group (Low dose group, LD), 13 mg/kg Ausome treated group (High dose group, HD) and non-treatment group (Ctrl). Each group contained 3 female rats and 3 male rats, which were weighing about 220 g or 330 g, to estimate the safety profiles in different sexes. Body weight measurements of animals were performed from day 0 (1 day before first injection of Ausome) to the end of the study (day 15), then serum and blood were isolated for serum biochemistry estimation and hematological analysis using an Automatic biochemical analyzer (TBA-40FR, Toshiba, Japan) and an Automated hematology analyzer (BC6600Plus, Mindray, Shenzhen, China). The major organs were isolated and sectioned for H&E staining to evaluate the histopathological change.

### Photothermal effects at the tumor region
4T1 breast tumor-bearing, 6–8 weeks old female Balb/c mice ($n = 3$ per group) were established as described above, and then intravenously injected with 15 mg/kg Ausome or AuNP. 6 h postinjection, the tumors were exposed to 660 nm laser with a power intensity of 1.2 W/cm³ for 30 min. The real time temperature changes at the tumor region as well as infrared thermal images of the mice were obtained using an infrared thermal camera (FLUK-TiS60 + 9 Hz, Fluke, USA).

To observe the influence of local hyperthermia on blood flow and vessel permeability, FITC-labeled dextran (70k Da) was intravenously injected (3 mg per mouse), immediately after 30-min laser irradiation. We carefully stripped the skin and exposed tumor tissue, the intratumoral FITC signals were detected using laser scanning confocal microscopy (ARsiMP-LSM-Kit-Legend Elite-USX, Nikon, Japan), the 3D reconstructed images were obtained through Z-stack scanning.

The thermally induced blood perfusion in the tumors was detected using real time multispectral optoacoustic tomography (MSOT). 4T1 breast cancer cells were inoculated into the axilla of 6–8 weeks-old female Balb/c mice ($n = 3$ per group). When the tumor size reached about 800 mm³, the mice were injected with 15 mg/kg Ausome or AuNP and treated with 660 nm laser irradiation, according to the procedures described above. The mice were then anesthetized with isoflurane and placed in a supine position in an animal holder, which was then inserted into an imaging chamber. Cross-sectional multispectral optoacoustic signals of tumors excited by an optical parametric oscillator laser at wavelengths from 690 to 900 nm in 10-nm steps were acquired and processed using an MSOT device (Invision 128, iThera, Germany).

### Immune contexture of tumor tissue
4T1 breast cancer cells were inoculated in situ at the left abdominal mammary fat pad of 6–8 weeks-old female Balb/c mice. After the tumor grew to about 100 mm³, the mice were divided into 4 groups: Ausome with laser irradiation (ASL), Ausome without laser irradiation (AS), AuNP with laser irradiation (ANL) and untreated control (Ctrl). 15 mg/kg Ausome or AuNP were intravenously injected on days 1 and 7. 6 h postinjection, the tumors were exposed to a 660 nm laser with a power intensity of 1.2 W/cm² for 30 min. After irradiation for 6 h, the tumors were isolated and weighted. To evaluate the levels of intratumoral cytokines and chemokines, the tissues were placed into cryotubes and homogenized in RIPA lysis solution supplemented with phenylmethanesulfonyl fluoride (PMSF) using a Tissue Lyser (Xinyi-48N, Xinyi, China) by shaking with sterile steel balls at 4 °C for 60 s. Afterwards, the soluble proteins were collected in the supernatant after centrifugation at 12,000× $g$ for 15 min at 4 °C. The concentrations of intratumoral cytokines and chemokines were detected using LEGENDPlex bead-based multiplex immunoassays kits and flow cytometry (Accuri™ C6, BD, USA) according to the manufacturer's instructions.

To study Ausome-mediated regulation of immune cell composition, tumors were resected at 6 or 24 h after the second laser irradiation on day 7. The tissues were cut into small pieces and treated with 2 mL Gey's Balanced Salt Solution (GBSS), supplemented with 1.0 mg/mL collagenase IV, 0.1 mg/mL DNase and 0.1 mg/mL hyaluronidase. After digesting at 37 °C for 20 min, the tissue solutions were ground against a 70 μm cell strainer, followed by centrifuging at 300 ×$g$ for 5 min to collect the cells. After fixation, the cells were incubated with fluorescent molecule-modified antibodies at 4 °C for 20 min to specifically label the various immune cells. To conjugate intracellular proteins, the cells were treated with permeabilization buffer prior to the antibody reaction. Next, the fluorescence intensities of the cells were measured by flow cytometry (Accuri™ C6, BD, USA). Freshly dissected tumor tissues were also fixed with 4% paraformaldehyde and made into paraffin-embedded tissue blocks, which were then sliced into 5–8 μm sections and immunohistochemistry stained with antibodies against CD4, CD8 or CD49, the positive cells were examined by light microscopy.

### Ausome triggered lymphocytes expansion in vivo
Ovalbumin expressing OVA breast cancer cells 4T1 (4T1-OVA) were in situ inoculated at the left abdominal mammary fat pad of 6–8 weeks-old female Balb/c mice ($n = 6$ per group). When the tumor grew to about 100 mm³, 15 mg/kg Ausome was intravenously injected 2 times on day 1 and day 7. 24 h post the last treatment, peripheral blood mononuclear cells (PBMC) from mice were isolated using Ficoll density gradient centrifugation. These cells were then seeded in 24-well plates and re-stimulated with 10 mg/L $OVA_{257-264}$ or $OVA_{323-339}$ peptide in RPMI 1640 medium containing GolgiPlug (BD, catalogue no: 555028), for 5 h to detect the activation or proliferation of antigen-specific CD4⁺ T and CD8⁺ T cells induced by Ausome. After incubation, the cells were transferred into 1.5 mL sterile centrifuge tubes and stained with fluorescence labeled antibodies against CD3, CD4, CD8 and IFN-γ. The frequencies of IFN-γ positive cells were analyzed using Acurri flow cytometry (C6, BD, USA). To examine Ausome-induced NK cell expansion, PBMCs were incubated with antibody against surface marker CD49 and analyzed by flow cytometry (AccuriTM C6, BD, USA).

### Local hyperthermia promoted infiltration of effector lymphocytes
6–8 weeks-old female Balb/c mice ($n = 6$ per group) were inoculated with 4T1-OVA breast cancer cells at the left abdominal mammary fat pad. When the tumor grew to about 100 mm³, mice were treated with ANL, AS or ASL on day 1 and day 7, according to the procedures described above. 2 h before the laser irradiation on day 7, all mice were

intravenously injected with $1 \times 10^5$ Tumor cells, which were isolated from the spleen of transgenic mice with MHCI restricted and OVA$_{257-264}$ specific TCR populations and subsequently labelled with CFSE according to the instructor manufacturer's instructions. After laser irradiated for 30 min, the tumor tissues were dissected to single cell suspensions to detect the proportion of CFSE-positive cells using a flow cytometry (AccuriTM C6, BD, USA).

### Cancer therapy experiments

To assess whether the immune modulation resulted in tumor inhibition, an orthotopic mammary cancer model and a subcutaneous colon cancer model were established. In brief, 4T1 breast cancer cells were injected into the mammary fat pad of 6–8 weeks old female Balb/c mice ($n = 6$ per group). When the tumors reached 50–100 mm$^3$, 3 therapeutic regimens, ASL, AS or ANL, were carried out. For ASL or ANL treatment, 15 mg/kg Ausome or AuNP were intravenously injected 3 times at a 3-day interval. 6 h post injection, the tumors were irradiated using a 660 nm laser with a power intensity of 1.2 W/cm$^2$ for 30 min. In the AS group, the mice were injected with 15 mg/kg Ausome without laser irradiation. Tumor size was monitored every other day using a caliper. Tumor volumes were calculated according to the formula (2):

$$V = 0.5 \times L \times S^2 \tag{2}$$

where L is the longer axis and S is the smaller axis.

CT26 colon cancer cells were subcutaneously inoculated into the right back of 6–8 weeks old female Balb/c mice ($n = 6$ per group). When the tumors reached 50–100 mm$^3$, the mice were divided into 4 groups (ASL, AS, ANL and untreated control) and treated according to the procedures described above. The tumor volumes were measured and calculated every other day.

### Ausome-mediated modulation of tumor sensitivity to PD-1 blockade therapy

An orthotopic breast cancer model was established in 6–8 weeks-old female Balb/c mice ($n = 6$ per group) as described above. The tumors were allowed to grow to 50–100 mm$^3$, at which time the mice were divided into 4 groups and treated with ASL, a PD-1 blockade antibody (Ab) or a combination of ASL and Ab (ASL + Ab). There was also an untreated control. For ASL treatment, Ausome was intravenously injected on days 1, 4 and 7, followed by 660 nm laser irradiation (1.2 W/cm$^2$, 30 min) at 6 h postadministration. The PD-1 blockade regimen was provided by intraperitoneal injection of a PD-1 blockade antibody (7.5 mg/kg) at 2-day intervals, for 5 total treatments. For the combinational therapy, ASL was performed on days 1, 4 and 7, and the antibody was administrated on days 1, 3, 5, 7 and 9. The tumor volume was measured every other day. The mice were euthanized on day 16, at which time tumor tissue samples were dissociated into single cells and stained with fluorescence-labeled antibodies against CD3, CD4 or CD8, and detected by flow cytometry to investigate the intratumoral T cell content after the 3 different treatments. The remaining tumor tissues were frozen and sectioned into 5–8 µm slices, followed by immunofluorescence staining with an antibody against PD-L1, as well as DAPI to label cell nuclei. Afterwards, the tissue sections were imaged using a confocal microscope (Zeiss 710, Zeiss, Germany).

### Ausome-mediated modulation enhancement of tumor sensitivity to chemotherapy

Orthotopic breast cancer bearing 6–8 weeks-old female Balb/c mice ($n = 6$ per group) were established following the procedures described above, and treated with ASL. Before laser irradiation, 1.5 mg/kg doxorubicin (Dox) was intravenously injected. To measure Dox-mediated immunogenic cell death (ICD), cell surface calreticulin (CRT) and intratumoral high mobility group box 1 (HMGB1) levels were assessed. ASL and Dox monotherapy were also carried out as controls. 24 h postirradiation, the tumor tissues were processed into single cells and stained with an Alexa Fluor 488-conjugated antibody against CRT. The fluorescence intensity on the tumor cells was then analyzed by flow cytometry (Accuri™ C6, BD, USA). In addition, protein was extracted from the tumor tissue as described above and the concentration of HMGB1 was detected using an ELISA kit following the manufacturer's instructions.

To assess the therapeutic effects of combined ASL and Dox treatment, as well as the ASL and Dox monotherapy, the materials were provided to 4T1 breast cancer-bearing mice on days 1, 4 and 7. The tumor size was measured every other day. The mice were euthanized on day 14, at which time the tumors were isolated and dissociated into single cells and stained with fluorescence-labeled antibodies against CD3, CD4 or CD8. Next, the intratumoral T cell content was detected by flow cytometry (Accuri™ C6, BD, USA). The freshly isolated tumor tissues were also frozen and sectioned into 5–8 µm slices, followed by immunofluorescence staining using antibodies against CD4 and CD8, as well as DAPI to label cell nuclei. Afterwards, the tissue sections were imaged using a confocal microscope (Zeiss 710, Zeiss, Germany).

### Statistical analysis

Quantitative results are presented as means ± standard deviation (s.d.), sample sizes (n) were empirically set at $n = 6$ for therapeutic studies and $n = 3–6$ for in vitro experiments or in vivo exploration of photothermal effects or immune responses, which are also specified in the figure legends. The analysis of RNA-seq data were performed using statistical software DESeq2 1.24.0. The statistical significances were determined by a two-tailed t-test, with the Benjamini-Hochberg method to correct P-values. Results were considered with significant differences at P values < 0.05. All other statistical analyses were performed using GraphPad Prism 8.4.2 (GraphPad Software, San Diego, USA). The statistical differences between two groups were determined by a two-tailed, Student's t-test. For multiple comparisons, one-way analysis of variance (ANOVA) followed by the Bonferroni multiple comparison was performed. Significant differences in tumor growth curves among multiple groups were analyzed by two-way ANOVA with the Bonferroni multiple comparison post-test. P values ≤ 0.05 were considered statistically significant (*$p < 0.05$, **$p < 0.01$, ***$p < 0.001$, ****$p < 0.0001$).

### Reporting summary

Further information on research design is available in the Nature Portfolio Reporting Summary linked to this article.

## Data availability

The RNA-Seq data generated in this study have been deposited in the NCBI database under accession code PRJNA978837. The mass spectrometry proteomics data have been deposited to the ProteomeXchange Consortium via the PRIDE partner repository with the dataset identifier PXD044359. The gating strategy for flow cytometry experiments can be found in Supplementary Figs. 25–31. The remaining data supporting the results of this study are available within the Article, Supplementary Information or Source Data file. Source data are provided with this paper.

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

## Acknowledgements

This work was supported by grants from the Beijing Natural Science Foundation of China (Z210017 to Y.L.), the National Basic Research Plan of China (2021YFA0909900 to R.Z.), the Strategic Priority Research Program of the Chinese Academy of Science (XDB36000000 to G.N.), the Key Area R&D Program of Guangdong Province (2020B0101020004 to R.Z.), CAS Project for Young Scientists in Basic Research (YSBR-041 to R.Z.) and the National Natural Science Foundation of China (82272953 to H.Q.).

## Author contributions

R.Z. and H.Q. conceived the study and designed the experiments. H.Q., Y.C., Z.W., N.L., Q.S., Y. Lin., W.Q., Y.Q. and L.C. performed the experiments. H.Q., Y.C. and R.Z. collected and analyzed the data. H.C., Y. Li and S.J. provided suggestions and technical support for the project. H.Q., R.Z., C.Y. and G.N. wrote the manuscript. G.N. supervised the project. All authors discussed the results and commented on the manuscript.

## Competing interests

The authors G.N., R.Z. and H.Q. filed a patent based on Ausome and its application in tumor therapy (Patent application number: CN202310327618.8. Title: "Bacterial mineralized metal nanoparticles for tumor therapy". The patent is currently submitted). The remaining authors declare no competing interests.
