## [Peer Review File · Nature Communications]

Biosynthesized gold nanoparticles that activate TLRs and elicit localized light-converting hyperthermia for pleiotropic tumor immunoregulationREVIEWER COMMENTS

Reviewer #1 (Remarks to the Author): with expertise in cancer immunology/immunotherapy
Review of 394889

This manuscript is directed towards the development of Ausome – a bacterial product formulated into a gold based nanoparticle as a novel platform to induce a heat inducible TLR dependent inflammatory response in solid tumors. The authors first demonstrate its production from E coli with compelling TEM and SEM dataset, and follow this with molecular characterization including its coating of outer leaflet of bacterial membrane. For their translational potential, they demonstrated its stability freeze/thaw. Rigorous characterization of ausome is compelling in Fig 1 and 2. However, the in vitro DC assay is still somewhat limited. They should show that the activated BMDC are activate T cells after mixing with ausome. With regards to the safety concerns, this should be addressed through monkey studies. While this is not currently recommended, there appears to be no gross toxicity in mice. An important concern is the negative control used throughout the paper. They used a Au-NP that was synthesized differently from the Ausome, which was produced from E coli, and purified. While their thermal effect appear to be the same, should they not use control Au particle that is made like Ausome, but whose particle surface is devoid of bacterial membrane. The PI suggest that this NP has a multi-layered MOA (heat generated inflammation, enhanced DC priming, increased vasculature), but the data presented are primarily of correlative nature, and the mechanistics studies are not fully fleshed out. Hence MOA of this platform is thin.

Outside of superficial lesions like skin, how would this be translated for visceral tumors?
While the innovation is high, translational potential is also thin.

Major issues:

Figure 2 – BMDC should be tested for T cell activation – surrogate markers of DC activation and gene expression profile is insufficient to show Ausome can prime T cells.

Fig 3 – the Ausome accumulation dataset in Fig 3c/d – what does n=3 mean? Is the figure representative from 3 experiments or did they use only 3 mice per group. If the former, how did they generate the errors? Also, the lack of tumor targeting is a concern – how does it accumulate in other tissue? This should be brought into the main figures.

Fig 4 – where is the control of laser only without any Au particles, either ausome or AuNP?

Reviewer #2 (Remarks to the Author): with expertise in nanotechnology

Gold nanoparticles have been widely used in catalysis, biomedicine, diagnostic and tumor therapy due to their excellent physical and chemical properties. Chemical synthesis is still the most used method to prepare gold nanoparticles, while the reaction conditions are harsh and additional modification is needed to increase the stability of gold nanoparticles. Minerals synthesized by macromolecules or microorganisms is an environmentally friendly method with mild conditions and no participation of toxicity chemicals, which can be regarded as 'green synthesis'. Microorganism driven biomineralization has been explored for a long time in the fields of noble metal recovery and environmental governance. However, the biomineralized products often show strong immunogenicity due to the attachment of microorganism related components.

The authors proposed a multi-layered immune regulation to remodel the tumor immune microenvironment, by making full use of the intrinsic and unique features of bacterium biomineralized gold nanoparticle. The diverse bacterial derived components triggered comprehensive systemic immune response, which was then selectively amplified in tumor area via gold nanoparticle mediated photo-thermal effects. With promoted infiltration of effector immune cells and other pro-inflammatory molecules in tumor tissue, the tumor immune microenvironment was improved, which facilitated local anti-tumor immunity and enhanced the efficacy of clinical first-line therapies, including chemotherapy and immune checkpoint inhibitors. To the best of my knowledge, this is the first work to explore the immune modulation potentials of bacterial generated gold nanoparticle, which brings new sights in the application of biomineralized products. The topic and results are very interesting. The manuscript should be accepted after minor revision. There are some problems needed to be addressed as follows.

1. Whether all the gold nanoparticles produced by *E. coli* secreted into the extracellular space; if not, what is the recovery rate of the biomineralized products, separated and collected following the method described in this paper; are there any approach to further increase the recovery rate?
2. How about the thermal stability of Ausome? Will the laser irradiation alter the physical and chemical properties of Ausome (surface charge, size).
3. Will the laser irradiation affect the immune stimulating efficacy of Ausome?
4. Some bacteria derived molecules, such as LPS, can affect the function or proliferation of cells; will Ausome be taken up by tumor cells and have a direct killing effect? After intravenous injection, will Ausome effect on vascular endothelial cells and cause toxicity?
5. In Figure 1b, c, gold nanoparticles should be marked.
6. In Figure 4d-f, representative positive cells should be marked.

Reviewer #3 (Remarks to the Author): with expertise in nanotechnology

In this paper, Nie et al developed an easily accessible bacterial biomineralization-generated immunomodulator and photothermal initiator, “Ausome” (Au+ [exo]some), which comprises a gold nanoparticle core with inducible hyperthermia effect covered by bacterial components with mobilization of diverse immune responses. Upon laser irradiation, tumor-accumulated Ausome could elicit a hyperthermic response, which improves tissue blood perfusion and contributes to enhanced infiltration of immunostimulatory modules, including cytokines and effector lymphocytes. Interestingly, the fancy word of “ausome” reminds me of “awesome”! Overall, the topic covered by the manuscript is certainly of significant interest to the readers of NC, and meet the criteria of this journal. This paper was well prepared, and the conclusions are well supported by the experimental results. Therefore, I am happy to recommend its acceptance for publication after minor revisions according to the following suggestions.

1. The authors mentioned that “although various interventions, including bacteria-derived agonists, cytokines, immune checkpoint blockade antibodies, have been developed to regulate the tumor microenvironment to augment immunity...” in the introduction, it is better to specify one or two examples briefly to let the audience get aware of relevant examples.
2. “Hence, an idea immune modulator for optimal antitumor efficacy, or for the augmentation of immunity, mobilizes diverse immune responses and selectively functions at the tumor site”, this sentence in line 57 is confusing, it is better to rewrite it.
3. A master scheme should be provided as Scheme 1 after the introduction, so that the entire story is more schematically clear to the readers.
4. In vitro photothermal cytotoxicity of Ausome against 4T1 cells should be provided to exhibit the antitumor effects in vitro.
5. How about the hydrodynamic and morphological stability of Ausome?
6. The resolution and quality of some of the figures should be checked carefully and improved, the fonts in figures are not clear and need to be enlarged.
7. The herein described “Ausome” reminds me of the recently reported bacteria-mimetic gold nanoparticles, which are gold nanoparticles coated with OMVs (Science Advances, 2022, 8, eabn1805). These bacteria-mimetic nanoparticles play exactly the same role, with photothermal properties and eliciting immune response. The authors are suggested to cite and discuss about this work, and comment on the differences. In a similar sense, the reviewer believes that the “Ausome” is likely also transported by macrophages to the tumor, relevant discussions should be added.

POINT-BY-POINT RESPONSE TO THE REVIEWER'S COMMENTS

We thank the reviewers for their constructive and valuable comments. Below the reviewers' comments are shown in blue bold font and our responses are in standard typeface. Sections showed in red font represent wording or data that has been altered/inserted into the revised manuscript. Other than the Figures in the revised main text and supplementary file, additional figures presented in this response letter were labeled as Figure Rs.

REVIEWER COMMENTS:

Reviewer #1: This manuscript is directed towards the development of Ausome – a bacterial product formulated into a gold based nanoparticle as a novel platform to induce a heat inducible TLR dependent inflammatory response in solid tumors. The authors first demonstrate its production from E coli with compelling TEM and SEM dataset, and follow this with molecular characterization including its coating of outer leaflet of bacterial membrane. For their translational potential, they demonstrated its stability freeze/thaw. Rigorous characterization of ausome is compelling in Fig 1 and 2.

1. However, the in vitro DC assay is still somewhat limited. They should show that the activated BMDC are activate T cells after mixing with ausome.

Our response 1: We thank the Reviewer for raising this issue. As suggested by the Reviewer, we carried out analysis on T cell activation after mixing with Ausome-treated BMDCs. Our new data reveal significantly enhanced T cell activation after incubating with pre-activated BMDCs. The latest data have been included in **Figure 2o** of the revised manuscript. The following sentences have been included in the main text.

Activated DCs then induced T cell responses either by surface co-stimulatory molecules and T-cell-receptor ligands or by secreted proinflammatory cytokines and growth factors, to modulate adaptive immune reactions³¹. Therefore, the activation of T cells by Ausome stimulated DCs was assessed. BMDCs were treated with Ausome for 12 h and then incubated with lymphocytes at a ratio of 1:5. After 24 h treatment, significantly increased proportions of CD4⁺ and CD8⁺ T cells with highly expressed CD69 or CD25 molecules were detected using flow cytometry (**Fig. 2o**). CD69 and

CD25 are the hallmarks of T cell activation and clonal expansion³², indicating the successful triggering of T cell responses by Ausome activated DCs.

Figure 20. Frequencies of CD69 or CD25 highly expressed CD4⁺ and CD8⁺ T cells after incubation with Ausome activated BMDCs for 24 h and detected using flow cytometry (n = 6 biologically independent samples).

2. With regards to the safety concerns, this should be addressed through monkey studies. While this is not currently recommended, there appears to be no gross toxicity in mice.

Our response 2: As suggested by the reviewer, safety is the most pressing issue concerning the clinical application of therapeutic agents. Here, we supplemented the determination of hematological parameters and morphological and functional assessment of major organs of rats treated with Ausome to further confirm the *in vivo* safety at therapeutic or higher doses. The new data have been included in **Supplementary Figure 12a-g and Supplementary Table 3** in the supplementary information, and the following descriptions have been added to the main text or supplementary information.

We then used female and male Sprague Dawley rats to validate the safety of Ausome when applied *in vivo* and to both female and male animals. A low dose of 6.5 mg/kg, and a high dose of 13 mg/kg, were tested through intravenous injection three times at a 3-day interval (**Supplementary Fig. 13a**). Compared to the non-treatment group, no significant difference in body weight was observed in both female and male rats during the experimental period (**Supplementary Fig. 13b, c**). Eight days after the third Ausome injection, the hematological parameters, serum biochemistry and histopathological estimation of major organs were measured. Increased amount of

platelets was observed in Ausome treated rats, both female and male, but within normal range even in high-dose treatment³⁶ (**Supplementary Table 3**). For other parameters, including white blood cells count and ratios, amount and features of red blood cells, some minor fluctuations were found among the three groups in both female and male rats, but no statistic differences were observed (**Supplementary Table 3**). The biochemistry analysis showed little serum indicator levels fluctuation after Ausome treatment in both female and male rats (**Supplementary Fig. 13d, e**). The major organs, including the brain, heart, lung, liver, spleen, kidney, and intestine, were also isolated and evaluated for safety profile of Ausome at the tested doses. We observed normal cell arrangements and detailed structures in all sectioned and stained organs, and no inflammatory or pathological changes were found (**Supplementary Fig. 13f, g**). In conclusion, Ausome elicited limited morphological and/or functional alterations in both female and male rats.

Supplementary Figure 12. Evaluation of Ausome biosafety profile in rats. a, Schematic illustration of experimental design of safety evaluation of Ausome in rats. Sprague-Dawley rats were divided into 3 groups, including non-treatment control group (Ctrl), low-dose treatment group (6.5 mg/kg Ausome, LD) and high-dose treatment

group (13 mg/kg Ausome, HD). And each group contained 3 female and 3 male rats. Ausome was intravenously injected for three times on days 1, 4 and 7. **b,c**, Body weights change of female rats (**b**) and male rats (**c**) throughout the study. **d,e**, Blood biochemistry analysis of ALT, AST, BUN and CREA in serum of female rats (**d**) and male rats (**e**, n = 3 biological independent samples) on day 15. **f,g**, Representative images of H&E stained organs sections isolated from female rats (**f**) and male rats (**g**) on day 16. Scale bars, 200 μ m. The numerical data in (**b-e**) are presented as the mean \pm s.d.

Supplementary Table 3. Hematological parameters estimations in rats treated with Ausome at the termination of study.

Parameters	Groups (Female)			Groups (Male)		
	Ctrl	LD	HD	Ctrl	LD	HD
WBC ($\times 10^9/L$)	2.51 \pm 1.02	2.76 \pm 0.97	1.93 \pm 0.79	1.41 \pm 0.26	2.30 \pm 0.67	1.97 \pm 0.79
Neu%	12.97 \pm 5.03	12.40 \pm 2.40	11.23 \pm 0.53	12.30 \pm 1.28	16.87 \pm 2.62	13.47 \pm 4.00
Lym%	83.80 \pm 7.17	85.20 \pm 2.00	85.10 \pm 1.28	83.97 \pm 3.38	78.63 \pm 2.26	80.73 \pm 7.07
Mon%	2.37 \pm 1.84	2.93 \pm 0.78	2.90 \pm 0.93	3.17 \pm 2.17	4.13 \pm 0.54	5.30 \pm 4.38
Eos%	0.60 \pm 0.43	0.10 \pm 0.08	0.33 \pm 0.21	0.23 \pm 0.33	0.20 \pm 0.08	0.13 \pm 0.05
Bas%	0.27 \pm 0.09	0.20 \pm 0.08	0.43 \pm 0.19	0.33 \pm 0.12	0.17 \pm 0.12	0.37 \pm 0.25
RBC ($\times 10^{12}/L$)	6.54 \pm 0.30	6.56 \pm 0.15	6.69 \pm 0.22	6.81 \pm 0.41	7.09 \pm 0.13	6.56 \pm 0.29
HGB (g/L)	133.00 \pm 5.89	133.33 \pm 0.47	136.00 \pm 4.32	142.67 \pm 6.55	144.33 \pm 1.70	133.33 \pm 4.64
HCT (%)	36.43 \pm 1.86	36.87 \pm 0.74	36.70 \pm 0.65	39.33 \pm 1.92	40.63 \pm 0.91	36.43 \pm 1.25
MCV (fL)	55.70 \pm 0.86	56.23 \pm 0.48	56.40 \pm 0.73	57.80 \pm 1.20	57.23 \pm 0.19	55.57 \pm 1.11
MCH (pg)	20.37 \pm 0.25	20.37 \pm 0.50	20.30 \pm 0.00	21.00 \pm 0.37	20.40 \pm 0.24	20.33 \pm 0.45
MCHC (g/L)	365.33 \pm 2.62	362.00 \pm 7.79	360.33 \pm 4.50	362.67 \pm 2.87	356.33 \pm 5.44	365.67 \pm 1.25
RDW-CV (%)	13.20 \pm 0.08	13.70 \pm 0.43	14.13 \pm 1.05	14.30 \pm 1.49	16.07 \pm 1.46	14.03 \pm 0.26

RDW-SD	29.13 ±	30.33 ±	31.50 ±	32.60 ±	36.40 ±	30.63 ±
(fL)	0.48	1.24	2.76	2.84	3.61	1.03
PLT (×	698.67 ±	717.33 ±	752.00 ±	759.33 ±	789.67 ±	835.33 ±
10 ⁹ /L)	79.37	129.91	22.55	28.6395	6.18	29.68
MPV (fL)	5.67 ±	5.83 ±	5.63 ±	5.77 ±	6.10 ±	5.97 ±
	0.12	0.36	0.17	0.37	0.28	0.12
PDW (fL)	6.17 ±	6.37 ±	6.03 ±	6.33 ±	6.83 ±	6.50 ±
	0.05	0.62	0.24	0.58	0.58	0.28
PCT (%)	0.40 ±	0.42 ±	0.42 ±	0.44 ±	0.48 ±	0.50 ±
	0.04	0.08	0.00	0.03	0.02	0.03

Data are presented as mean ± s.d. (n = 3). WBC, White blood cell counts; Neu, Neutrophils; Lym, Lymphocytes; Mon, Monocytes; Eos, Eosinophils; Bas, Basophils; RBC, Red blood cell counts; HGB, Hemoglobin; HCT, Hematocrit; MCV, Mean corpuscular volume; MCH, Mean corpuscular hemoglobin; MCHC, Mean corpuscular hemoglobin concentration; RDW-CV, Red cell distribution width-coefficient variation; RDW-SD, Red cell distribution width-standard deviation; PLT, Platelets; MPV, Mean platelet volume; PDW, Platelet distribution width; PCT, Plateletcrit.

3. An important concern is the negative control used throughout the paper. They used a Au-NP that was synthesized differently from the Ausome, which was produced from E coli, and purified. While their thermal effect appear to be the same, should they not use control Au particle that is made like Ausome, but whose particle surface is devoid of bacterial membrane.

Our response 3: We appreciate the Reviewer’s advice, which may make this research more rigorous. We have taken this suggestion and tried to prepare gold nanoparticles (negative control) by stripping the bacterial molecules attached on Ausome.

The interactions between biomacromolecules and gold nanoparticles are based on weak covalent binding of sulfhydryl or amino group to Au and other intermolecular forces, in which Au-S bond is the strongest. Therefore, we first attempted to utilize dithiothreitol and glutathione, two small molecules with sulfhydryl groups, to competitively bind gold nanoparticles and release the original conjugated bacterial components. However, as shown in Figure R1, after adding dithiothreitol (B) or glutathione (C), insoluble matters precipitated out of the solution. Transmission electron microscope (TEM) images also observed these aggregates. The aggregated

nanoparticles could be greatly different with Ausome in *in vivo* circulation and photothermal features, thus these products are not proper to be used as negative control of Ausome. We then tried the thermal decomposition of organic molecules on Ausome by calcining at 600 °C for 1 h using a tube furnace, and the residua were collected and redissolved in PBS buffer (D) or sodium citrate solution (E). The redissolved solution did not return to uniformly dispersive, red gold nanoparticle solution, as Ausome solution shown in Figure R1A, black blocks suspended in both PBS and sodium citrate solutions, no nanoparticles were observed using TEM. In conclusion, we have tried some approaches, but all failed to remove the bacterial molecules from Ausome without alternation of physicochemical properties at the same time.

Figure R1. Gold nanoparticle solutions (upper panels) and transmission electron microscopy images (lower panels) before (A) and after (B-E) removing bacterial components from Ausome using different approaches.

In addition, the critical control is pure heat treatment, similar to Ausome but without other influences. The synthesized PEGylation AuNP is easily accessible, with similar physicochemical properties and photothermal features as Ausome, while eliciting little immune reactions. We also tested the immunogenicity of the AuNP using bone-marrow dendritic cells (BMDCs) as a model. As shown in **Figure 9c, d, supplementary information**, no immune stimulating effect was detected. Therefore, using AuNP as a control can also demonstrate the thermal effects of Ausome. The new data have been included in **Supplementary Figure 9c, d**, and the following descriptions have been added to the main text or supplementary information.

The synthetic AuNP or single laser irradiation treatment without any gold nanoparticles failed to facilitate the differentiation of BMDCs to mature phenotype (**Supplementary Fig. 9c, d**)

Supplementary Figure 9. c, Representative flow cytometry dot plots of BMDCs (2×10^5 cells per sample) stained with fluorescence-labeled antibodies against CD80 and CD86, after treating with 30 min laser irradiation or AuNP. **d**, Statistics chart of CD80⁺CD86⁺ BMDCs after treating with 30 min laser irradiation, AuNP (2.5 $\mu\text{g/mL}$), Ausome (2.5 $\mu\text{g/mL}$) or irradiated Ausome (660 nm, 1.5 W/cm² laser irradiated for 30 min before incubating with BMDCs) for 24 h ($n = 3$ biological independent samples). The data are presented as the mean \pm s.d. **** $p < 0.0001$; significant differences were analyzed by one-way ANOVA followed by the Bonferroni multiple comparison test (**d**).

4. The PI suggest that this NP has a multi-layered MOA (heat generated inflammation, enhanced DC priming, increased vasculature), but the data presented are primarily of correlative nature, and the mechanistics studies are not fully fleshed out. Hence MOA of this platform is thin.

Our response 4: We have included additional data to present the immune responses in different stages of the multi-layered regulating process. The new data have been included in **Figure 4 d-g**, and the following descriptions have been added to the main text or supplementary information.

To better demonstrate the immune cell responses occurring at different stages in ASL-regulated immunity, we measured Ausome-triggered expansion of lymphocytes and local hyperthermia-promoted intratumoral infiltration respectively. As schemed in **Fig. 4d**, after intravenous injection, Ausome could rapidly be recognized by innate immune cells, which then mature to release pro-inflammatory cytokines (**Supplementary Fig. 8b, Fig. 11d, e, Fig. 18a-e**) and promoted effector immune cells activation and proliferation. To evaluate tumor-specific T-cell responses, we used a 4T1 breast cancer cell expressing ovalbumin (4T1-OVA) to establish the mouse tumor model. The levels of specific CD8⁺ and CD4⁺ T cells for MHC-I epitope OT-1 and

MHC-II epitope OT-2 of OVA in peripheral blood were measured then. Compared to the tumor-bearing mice in the non-treated control group, Ausome treatment markedly increased the tumor-specific T cells, with over 6 times more OT-1 specific CD8⁺ T cells detected (**Fig. 4e**). Significant elevation of another important immune population, NK cells, was also observed within Ausome treatment (**Fig. 4e**), indicating a comprehensive proliferation or expansion of effector immune cells. The promoted tumor infiltration of these circulating effector lymphocytes by Ausome-mediated local hyperthermia was investigated then. As presented in **Fig. 4f**, 4T1-OVA bearing mice were immunized with Ausome twice as described above. 2 h before the second irradiation, carboxyfluorescein succinimidyl ester (CFSE) labeled OT-1 T cells, which was isolated from a transgenic mouse with TCR populations that specifically recognized OT-1 bound to H-2K^b complex, were intravenously injected. The tumor infiltrated CFSE-positive cells were measured using flow cytometry. We observed improved OT-1 cell accumulation in tumor tissue with local hyperthermia treatment. At the same time, Ausome-mediated thermal therapy further facilitated this phenomenon (**Fig. 4g**), which suggested other mechanisms in hyperthermia promoted immune cell infiltration besides blood perfusion.

Figure 4. **d**, Schematic illustration of the procedure of Ausome activating effector lymphocytes. After encountering the intravenously injected Ausome, innate immune cells recognized them and matured to release a high amount of pro-inflammatory cytokines, which promoted the expansion of tumor-specific T cells or NK cells. **e**, Balb/c mice were *in situ* inoculated with 4T1-OVA breast tumor cells and treated with Ausome when the tumor grew to approximately 100 mm³, the OT-1 specific CD8⁺ T cells, OT-2 specific CD4⁺ T cells and NK cells in peripheral blood were analyzed using flow cytometry (n = 6 biologically independent experiments). **f**, Experimental design of local hyperthermia promoted effector lymphocyte infiltration in tumor tissue. 4T1-

OVA tumor-bearing Balb/c mice were treated with ANL, AS, or ASL twice on day 1 and day 7. 2 h before the second laser irradiation, CFSE labeled T cells that isolated from the spleen of a transgenic mouse with TCR specific for antigen epitope OT-1, were intravenously injected to mice. **g**, Representative flow cytometry dot plots and statistical chart of CFSE positive cells in tumors (n = 6 biologically independent experiments).

5. Outside of superficial lesions like skin, how would this be translated for visceral tumors? While the innovation is high, translational potential is also thin.

Our response 5: Thanks for the Reviewer's insightful comments and affirmation on the innovation of our work. The superficial tumor does have a natural advantage on laser dependent therapy, and the application of laser therapy to internal tumors has been a highly concerned issue.

Two laser devices, the Visualase Thermal Therapy System and the Neuroblate Laser Ablation System, have been approved by the FDA for tumor treatment; large amounts of clinical research on PTT have also been conducted in patients with brain cancer (Belykh. *Surg. Neurol. Int.* 2017; Hawasli. *Neurosurg. Focus.* 2014), liver cancer (Gough-Palmer. *World J. Gastroenterol.* 2008; Vogl. *Radiology.* 2004; Vogl. *Radiology.* 2002; Arienti. *Radiology.* 2008), prostate cancer (Wenger. *Curr. Opin. Urol.* 2014; Rastinehad, *PNAS*, 2019) or head and neck cancer (NCT00848042). In conclusion, interventional laser devices or technologies are readily achieved in clinics. Therefore, our task for preclinical research is to enable rapid, reproducible and less resource-intensive testing of a wide variety of novel compounds and materials in easily accessible animal tumor models. Although we are willing to demonstrate the application of ASL therapy in non-superficial tumors, the truth is that it's not easy for us to development a mouse-using interventional laser device, but based on the rapidly developed clinical PTT devices and technologies, we have faith in the translational application of Ausome in visceral or other non-superficial tumors.

Major issues:

6. Figure 2 – BMDC should be tested for T cell activation – surrogate markers of DC activation and gene expression profile is insufficient to show Ausome can

prime T cells.

Our response 6: Please refer to our response to Reviewer #1, question 1.

7. Fig 3 – the Ausome accumulation dataset in Fig 3c/d – what does n=3 mean? Is the figure representative from 3 experiments or did they use only 3 mice per group. If the former, how did they generate the errors?

Our response 7: We have used 3 mice per group in these experiments and labeled n = 3 biologically independent experiments in the figure legend of 3c and 3d. The figures in 3b are representative images, from randomly chosen one of the 3 mice.

8. Also, the lack of tumor targeting is a concern – how does it accumulate in other tissue? This should be brought into the main figures.

Our response 8: We appreciate the Reviewer for bringing this important issue to us. In fact, the Ausome does have some tumor-targeting and retention ability, as showed in **Figure 3b**, and thermal treatment could further increase this local accumulation, more than 4 times Ausome content was detected in tumor tissue after laser irradiation (**Figure 4c**). This thermal enhanced nanoparticle infiltration has also been proven in our previous work (Zhao. et al., *ACS Nano*, 2017). Nevertheless, we agree with the reviewer's opinion, enhancing the tumor targeting of Ausome inevitably brings more benefits on the ultimate therapeutic outcomes. There are various strategies could be utilized to promote the accumulation of Ausome to tumor tissue, including modifying targeting ligands on Ausome via genetic engineering or chemical conjugation, hitchhiking of phagocytic immune cell to inflammatory tumor tissue (Gao et al. *Sci. Adv.* 2022) or intratumoral injection of Ausome to achieve high levels of Ausome in local tumor, which have been complemented to the present manuscript as discussions; but these are more like another topic and we'd like to verify these proposals in our future work. The following descriptions have been added to the main text:

Through genetic engineering or chemical conjugation, Ausome could be modified with specific targeting ligands, to promote its accumulation in tumor tissue, thus enhancing the ultimate anti-tumor therapeutic efficacy and optimizing the laser irradiation conditions, such as lower laser power density. Gao et al. recently reported a

strategy of delivering *E. coli* OMV-coated gold nanoparticles to inflammatory tumor tissues by *in vivo* hitchhiking phagocytic immune cells, with ingenious design of intracellular self-assembly to minimize exocytosis of nanoparticles from immune cells³⁰. Gao's study exhibits another alternative for tumor targeted transport of Ausome, of which the clever design of gold nanoparticles aggregation could also lead to a red-shift of the maximum absorption, therefore, a longer wavelength laser with increased penetration depth could be used to inspire the photothermal effect of Ausome.

The accumulation of nanoparticle in organs besides tumor have also been reported in many other studies, no matter by intravenous injection or subcutaneous injection (Kumar et al., *Adv. Drug Deliv. Rev.*, 2023). With abundant bacteria derived components, which are natural and robust immune-stimulating agents, it is not unexpected to observe Ausome accumulated to livers, spleen, lymph nodes and lungs, all peripheral lymphatic tissues or organs with immune features. After intravenous injection, Ausome spread throughout the body along the blood flow, which could be recognized and taken up by circulating innate immune cells and brought to these immune-related tissues for downstream reactions or directly interacted with the tissue resident immune cells, both facilitated accumulation of Ausome. These *in vivo* behaviors of Ausome actually implied activated systemic immunity and further confirmed its immune stimulating ability.

In addition, we would like to keep those data as a supplementary information, because the topic of Figure 3 is to demonstrate Ausome mediated, laser inspired tumor local hyperthermia and consequent thermal effects, but biodistribution of Ausome reflected more on immune responses and supplied no further information on tumor accumulated Ausome or subsequent local hyperthermia. Moreover, there are strong evidences (**Figure 3b-e**) to show the intratumoral Ausome.

9. Fig 4 – where is the control of laser only without any Au particles, either ausome or AuNP?

Our response 9: We have demonstrated that under the irradiation conditions that Ausome or AuNP could heat the local temperature to 42°C, laser only treatment did not raise the temperature in local tumor (**Figure 4f, g**). There are no damages observed on the skin after laser irradiating on a superficial tumor (**Supplementary Figure 15**). In

the revised version, we further complemented data to illustrate that the laser alone could not exert significant effects on cell viability, immune activation, as well as tumor treatment outcomes. The new data have been included in **Supplementary Figure 9c, d, 10b and 21**, and the following descriptions have been added to the main text or supplementary information.

The synthetic AuNP or single laser irradiation treatment without any gold nanoparticles failed to facilitate the differentiation of BMDCs to mature phenotype (**Supplementary Fig. 9c, d**).

Supplementary Figure 9. c, Representative flow cytometry dot plots of BMDCs (2×10^5 cells per sample) stained with fluorescence labeled antibodies against CD80 and CD86, after treating with 30 min laser irradiation or AuNP. **d**, Statistics chart of CD80⁺CD86⁺ BMDCs after treating with 30 min laser irradiation, AuNP (2.5 $\mu\text{g}/\text{mL}$), Ausome (2.5 $\mu\text{g}/\text{mL}$) or irradiated Ausome (660 nm, 1.5 W/cm^2 laser irradiated for 30 min before incubating with BMDCs) for 24 h ($n = 3$ biological independent samples). The data are presented as the mean \pm s.d. **** $p < 0.0001$; significant differences were analyzed by one-way ANOVA followed by the Bonferroni multiple comparison test (**d**).

No difference in cell viability was observed in 4T1 cells with other treatments, including laser only, AuNP or irradiated Ausome (single components or different status of Ausome mediated therapeutic process, **Supplementary Fig. 10b**). These results imply that direct cell killing may not be the key mechanism of Ausome mediated anti-tumor therapy in this study.

Supplementary Figure 10. Cytotoxicity of Ausome on tumor cells. b, Viabilities of 4T1 cells after treating with 30 min laser irradiation, chemical synthetic AuNP (150 $\mu\text{g/mL}$), Ausome (150 $\mu\text{g/mL}$) or irradiated Ausome (660 nm, 1.5 W/cm^2 laser irradiated for 30 min before incubating with 4T1 cells) for 24 h ($n = 4$ biological independent samples). The data are presented as the mean \pm s.d.

To assess the therapeutic effects of hyperthermia alone, we applied AuNP following the same procedure described above. This strategy resulted in slightly controlled tumor growth, but without a significant difference when compared to the untreated group (**Fig. 4m**), while the laser alone elicited little suppression on tumor growth (**Supplementary Fig. 21**), indicating the slight tumor inhibition was the local hyperthermia resulted effects.

Supplementary Figure 21. Evaluation of the antitumor effects of laser irradiation.

Balb/c mice were *in situ* inoculated with 4T1 breast cancer cells. When the volume reached 50-100 mm^3 (day 0), the tumors were exposed to 660 nm laser (1.2 W/cm^2) for 30 min with (ASL group) or without (Laser irradiation group) previously administrated Ausome. Treatments were performed three times, and the average and individual tumor volumes of the 4T1 breast cancer models during the therapeutic experiment ($n = 6$ biologically independent animals) are shown. The numerical data are shown as the mean \pm s.d.

Reviewer #2: Gold nanoparticles have been widely used in catalysis, biomedicine, diagnostic and tumor therapy due to their excellent physical and chemical properties. Chemical synthesis is still the most used method to prepare gold nanoparticles, while the reaction conditions are harsh and additional modification is needed to increase the stability of gold nanoparticles. Minerals synthesized by macromolecules or microorganisms is an environmentally friendly method with mild conditions and no participation of toxicity chemicals, which can be regarded as ‘green synthesis’. Microorganism driven biomineralization has been explored for a long time in the fields of noble metal recovery and environmental governance. However, the biomineralized products often show strong immunogenicity due to the attachment of microorganism related components.

The authors proposed a multi-layered immune regulation to remodel the tumor immune microenvironment, by making full use of the intrinsic and unique features of bacterium biomineralized gold nanoparticle. The diverse bacterial derived components triggered comprehensive systemic immune response, which was then selectively amplified in tumor area via gold nanoparticle mediated photo-thermal effects. With promoted infiltration of effector immune cells and other pro-inflammatory molecules in tumor tissue, the tumor immune microenvironment was improved, which facilitated local anti-tumor immunity and enhanced the efficacy of clinical first-line therapies, including chemotherapy and immune checkpoint inhibitors. To the best of my knowledge, this is the first work to explore the immune modulation potentials of bacterial generated gold nanoparticle, which brings new sights in the application of biomineralized products. The topic and results are very interesting. The manuscript should be accepted after minor revision. There are some problems needed to be addressed as follows.

1. Whether all the gold nanoparticles produced by *E. coli* secreted into the extracellular space; if not, what is the recovery rate of the biomineralized products, separated and collected following the method described in this paper; are there any approach to further increase the recovery rate?

Our response 1: Thank you for raising this issue. The yield is indeed very critical for

large scale production. In the current study, we collect the products on the 5th day post introduction of chloroauric acid. After the first low-speed centrifugation, the bacteria were separated from the supernatant Ausome. We observed dark red sediment, other than white cell mass, indicating there are still a lot of gold nanoparticles conjugated on the bacterial cells and not yet being secreted to the extracellular space. The transmission electron microscopy image also showed attachment of gold nanoparticles on bacterial cells (Figure R2).

Figure R2. Transmission electron microscopy image of *E. coli* after incubating with chloroauric acid for 5 days. Red arrows indicated gold nanoparticles. Scale bar, 200 nm.

Following the production procedure described in the current study, the yield of biomineralized gold nanoparticles is about 10%, which is calculated according to the ICP-MS quantified Au element content in the ultimate products and the initial added chloroauric acid. This relatively low recovery rate reminds us there is still a lot of room for further improvement of the yield of Ausome, through some possible methods, such as facilitating the secretion of Ausome from bacteria or optimizing production process.

According to the experimental observation (**Figure 1d, e and Supplementary Figure 1**) and theoretical consideration (the embedding of nanosized particles in the periplasmic space led to localized pressure, which altered the curvature of the outer membrane and is a mechanism to initiate bacterial membrane vesicle genesis), we have proposed Ausome may be secreted to extracellular space in a membrane bubbling manner. Therefore, other than optimizing the production processes, increase in the genesis of membrane vesicle could also benefit a lot to the Ausome yield. The secretion of membrane vesicles is an important pathway for bacteria to defense extracellular stresses, including antibiotics (Kulp et al. *Annu. Rev. Microbiol.* 2010) or irradiation (Zhang et al. *Front. Microbiol.* 2020). We have performed additional experiments to expose bacterial suspensions to X-ray irradiation at dose of 1 kGy after *E. coli* was incubated with chloroauric acid for 5 days, and the Ausome were then collected and

quantified using ICP-MS. Approximately 2 times more Au content was detected with X-ray irradiation treatment, and scanning electron microscopy measurement showed wrinkled bacterial surface with large amounts of bubbles (Figure R3), indicating increased secretion of membrane vesicles, as well as promoted yield of Ausome. However, these stresses could have significant influences on the biological activities of bacteria and altered the composition of secreted membrane vesicles. Thus, stress generated additional Ausome may have different immune regulating effects compared to Ausome studied in the current study; therefore, a comprehensive evaluation on the *in vivo* safety and immune responses stimulated by these stress generated Ausome indeed needs further characterization and we would like to perform in-depth investigation in our future research.

Figure R3. Scanning electron microscopy image of *E. coli* with or without 1 kGy X-ray irradiation. Scale bar, 2 μ m.

2. How about the thermal stability of Ausome? Will the laser irradiation alter the physical and chemical properties of Ausome (surface charge, size).

Our response 2: This question is well taken. We have performed additional experiments to evaluate the morphology, size and zeta potential of Ausome after heating by laser irradiation. No significant alteration on these physicochemical properties was observed, indicating high thermal stability of Ausome. The data have been included in **Supplementary Figure 5a-c**, and the following descriptions have been added to the main text or supplementary information.

We also evaluated the influence of laser irradiation on the physicochemical features of Ausome. After exposing to laser for 30 min, Ausome was assessed using TEM and DLS, and little difference in morphology, diameter and zeta potential was observed when compared to fresh prepared Ausome (**Supplementary Fig. 5a-c**), revealing high laser stability of Ausome.

Supplementary Figure 5. Stability of Ausome under different storage conditions and laser irradiation. Representative TEM images (a), size distributions (b), and surface zeta potentials (c) of Ausome after repeated freezing and thawing (three cycles in one-week intervals, left), long time freezing (-80°C for 6 months) or laser irradiation (1.5 W/cm²) for 30 min. Scale bars, 200 nm.

3. Will the laser irradiation affect the immune stimulating efficacy of Ausome?

Our response 3: We have supplemented experiments and taken the ability of irradiated Ausome to trigger dendritic cells (DCs) mature as a reference to evaluate its immune stimulating efficacy. No significant difference on the frequencies of CD80⁺CD86⁺ DC cells was detected between Ausome and irradiated Ausome treatment, which revealed laser irradiation generating no effect on the immune stimulating efficacy of Ausome. The new data have been included in **Figure 2n and Supplementary Figure 9d**, and the following description have been supplemented to the main text or supplementary information.

We also investigated whether laser irradiation would affect the immune stimulating potency of Ausome. Ausome was exposed in laser (660 nm, 1.5W/cm²) for 30 min and then added to the medium of BMDCs. Significantly enhanced frequency of CD80⁺CD86⁺ BMDCs was detected after incubation for 24 h and no apparent differences in triggering DCs activation were observed between Ausome with or without laser irradiation (**Fig. 2n, Supplementary Fig. 9c, d**), indicating irradiation affected little on immune stimulating efficacy of Ausome.

Figure 2n, Representative flow cytometry dot plots of CD80⁺ and CD86⁺ BMDCs (2×10^5 cells per sample), after treatment with irradiated Ausome (660 nm, 1.5W/cm² laser irradiated for 30 min before incubating with BMDCs) at a concentration of 2.5 μ g/mL for 24 h (**n**).

Supplementary Figure 9. Ausome-induced DC maturation. d, Statistic analysis of CD80⁺CD86⁺ BMDCs after treating with 30-min laser irradiation, AuNP (2.5 μ g/mL), Ausome (2.5 μ g/mL) or irradiated Ausome (660 nm, 1.5 W/cm² laser irradiated for 30 min before incubating with BMDCs) for 24 h ($n = 3$ biological independent samples). The data are presented as the mean \pm s.d. **** $p < 0.0001$; significant differences were analyzed by one-way ANOVA followed by the Bonferroni multiple comparison test (**d**).

4. Some bacteria derived molecules, such as LPS, can affect the function or proliferation of cells; will Ausome be taken up by tumor cells and have a direct killing effect? After intravenous injection, will Ausome effect on vascular endothelial cells and cause toxicity?

Our response 4: We thank the Reviewer for raising this important issue. To address it, we carried out additional experiments to assess the uptake of Ausome by tumors cells and the cytotoxicity of Ausome on tumor cells, as well as vascular endothelial cells. We also investigated the structural integrity and arrangement of endothelial cells in blood vessels of heart, after intravenous injection of Ausome. Confocal images showed

Ausome could be taken up by tumor cells (Figure R4), while only a minor decrease in cell viability was detected in 4T1 breast tumor cells and vascular endothelial cells after Ausome treatment at a dose as high as 300 $\mu\text{g/mL}$ (**Supplementary Figure 10a, Figure 11b**). No obvious morphological or structural change was observed in heart sections immune-stained with CD31 to label endothelial cells (**Supplementary Figure 11c**). These data revealed that the direct tumor killing may not be the key mechanism of the Ausome mediated antitumor activity in the current study; and the Ausome generated no or low toxicity on vascular endothelial cells either *in vitro* or *in vivo*. The new data have been included in **Supplementary Figure 10a and 11b, c**, and the following description have been supplemented to the main text or supplementary information.

Figure R4. Two-photon luminescence (TPL) images of 4T1 tumor cells after incubation with AuNP or Ausome for 24 h. Quantification of Au taken up by 4T1 cells after incubating with AuNP or Ausome for 6 h or 24 h.

We next investigated whether Ausome could elicit direct cytotoxicity on tumor cells by incubating 4T1 breast cancer cells with Ausome. The cell viabilities were not affected by Ausome at concentrations up to 150 $\mu\text{g/mL}$, and increased dose or additional laser irradiation only elicited minor cytotoxicity (**Supplementary Fig. 10a**).

Supplementary Figure 10. Cytotoxicity of Ausome on tumor cells. a, 4T1 breast cancer cells were treated with Ausome at indicated doses. 24 h post these treatments, the cell viabilities were analyzed using Cell-Counting-Kit-8 assay (n = 4 biological independent samples).

Apart from the blood cells, vascular endothelial cells are also exposed to Ausome

and cytotoxicity of Ausome on vascular endothelial cells was evaluated then. Human umbilical vein endothelial cells (HUVECs) were incubated with Ausome at a concentration range of Ausome for 24 h, and no significant decrease in cell viability was observed, even in concentration as high as 300 $\mu\text{g/mL}$ (**Supplementary Fig. 11b**). We also assessed the arrangement and structure of endothelial cells *in vivo*. High-dose Ausome (270 μg per mouse) were intravenously injected to Balb/c mice. The hearts, with adequate blood supply and abundant blood vessels, were isolated 48 h post injection and immuno-stained to visualize vascular endothelial cells. No disturbance was observed in Ausome-treated blood vessels, endothelial cells maintained as narrow strip, which were closely connected together and arranged alongside the blood vessels, with intact structure and smooth surface (**Supplementary Fig. 11c**).

Supplementary Figure 11b, Cytotoxicity analysis of Ausome on vascular endothelial cells by treating human umbilical vein endothelial cells (HUVECs) with Ausome at different concentrations for 24 h, and cell viabilities were detected using Cell-Counting-Kit-8 assay (n = 4 biological independent samples). **c**, Heart sections from mice treated with high-dose Ausome (270 μg per mouse), which were then immune-stained with an antibody against CD31 to label vascular endothelial cells. Inserted boxes indicated high coverage. Scale bars, 50 μm and 200 μm (inserted images).

5. In Figure 1b, c, gold nanoparticles should be marked.

Our response 5: We have added red arrow to point out gold nanoparticles on bacteria cells, and revised figures have been presented in **Figure 1b, c**.

Fig. 1. Ausome production by *E. coli*. b,c, TEM (b) and SEM (c) images of *E. coli* and the generated gold nanoparticles at the indicated culture time points. Dashed boxes indicate the enlarged region. **The red arrows indicate the biom mineralized gold nanoparticles.** Scale bars: 1 μm (b, upper panels), 200 nm (b, lower panels), 4 μm (c, upper panels), 500 nm (c, lower panels).

6. In Figure 4d-f, representative positive cells should be marked.

Our response 6: We thank the Reviewer for pointing out this issue. We have added red arrow to mark the positive cells, revised figures have been presented in **Supplementary Figure 19.**

Supplementary Figure 19. Tumor T cell infiltration after Ausome-mediated, multi-layered modulation. **Representative immunohistochemical images of tumor sections isolated from mice treated with AS, ANL or ASL, T cells and NK cells with specific antibodies against CD4, CD8 and CD49, respectively. The red arrow indicated positive cells. Scale bar, 100 μm .**

Reviewer #3: In this paper, Nie et al developed an easily accessible bacterial biomineralization-generated immunomodulator and photothermal initiator, “Ausome” (Au+ [exo]some), which comprises a gold nanoparticle core with inducible hyperthermia effect covered by bacterial components with mobilization of diverse immune responses. Upon laser irradiation, tumor-accumulated Ausome could elicit a hyperthermic response, which improves tissue blood perfusion and contributes to enhanced infiltration of immunostimulatory modules, including cytokines and effector lymphocytes. Interestingly, the fancy word of “ausome” reminds me of “awesome”! Overall, the topic covered by the manuscript is certainly of significant interest to the readers of NC, and meet the criteria of this journal. This paper was well prepared, and the conclusions are well supported by the experimental results. Therefore, I am happy to recommend its acceptance for publication after minor revisions according to the following suggestions.

1. The authors mentioned that “although various interventions, including bacteria-derived agonists, cytokines, immune checkpoint blockade antibodies, have been developed to regulate the tumor microenvironment to augment immunity...” in the introduction, it is better to specify one or two examples briefly to let the audience get aware of relevant examples.

Our response 1: Thank you for reminding us this detailed information. We have added some examples to each intervention strategy, following is the revised description.

Although various interventions, including bacteria-derived agonists (e.g., MPLA, poly-ICLC)^{9,10}, cytokines (e.g., interleukin-2, interferon- α)¹¹, immune checkpoint blockade antibodies [e.g., antibodies against programmed cell death protein 1(PD-1) or its ligand PD-L1]¹², and vascular remodeling agents [e.g., antibodies vascular endothelial growth factor (VEGF) or angiopoietin 2 (ANG2)]¹³, have been developed and explored to regulate the TME to augment immunity, the therapeutic outcomes remain unsatisfactory.

2. “Hence, an idea immune modulator for optimal antitumor efficacy, or for the

augmentation of immunity, mobilizes diverse immune responses and selectively functions at the tumor site”, this sentence in line 57 is confusing, it is better to rewrite it.

Our response 2: Thanks for raising this issue. We have rewritten it and following is the revised sentence.

Hence, an ideal immunomodulator is to mobilize diverse immune responses and selectively functions at the tumor site, thus reaching augmented immunity and enhanced antitumor efficacy.

3. A master scheme should be provided as Scheme 1 after the introduction, so that the entire story is more schematically clear to the readers.

Our response 3: Thanks for the reviewer’s advice. As suggested, we have moved the overall scheme to the end of the introduction section.

4. In vitro photothermal cytotoxicity of Ausome against 4T1 cells should be provided to exhibit the antitumor effects in vitro.

Our response 4: This is a well taken suggestion. We have performed additional experiments to exhibit the antitumor effects of Ausome through a CCK-8 assay. Cell viabilities of 4T1 cells was detected after treating with a series dose of Ausome or Ausome combined with thermal treatment (generated by laser irradiation). No significant difference in cell viability was observed even in high-dose Ausome treated group, and additional mild hyperthermia treatment didn’t generate further cytotoxicity on 4T1 tumor cells, which may reveal that direct tumor killing is not the key mechanism of Ausome-mediated antitumor therapy in the current study. The new data have been included in **Supplementary Figure 10a**, and the following description have been supplemented to the main text or supplementary information.

We next investigated whether Ausome could elicit direct cytotoxicity on tumor cells by incubating 4T1 breast cancer cells with Ausome. The cell viabilities were not affected by Ausome at concentrations up to 150 $\mu\text{g/mL}$, and increased dose or additional laser irradiation only elicited minor cytotoxicity (**Supplementary Fig. 10a**).

Supplementary Figure 10. Cytotoxicity of Ausome on tumor cells. a, 4T1 breast cancer cells were treated with Ausome at indicated doses (left histogram) or with Ausome and additional 30 min laser irradiation (right histogram). 24 h post these treatments, the cell viabilities were analyzed using Cell-Counting-Kit-8 assay (n = 4 biological independent samples).

5. How about the hydrodynamic and morphological stability of Ausome?

Our response 5: We have carried out additional experiments to evaluate the hydrodynamic and morphological stability of Ausome in conditions including repeated freezing-thawing, long-term freezing, as well as laser irradiation. No obvious aggregation or alternation of size or micro-morphology was detected, indicating great stability of Ausome. Related data have been included in **Supplementary Figure 5a-c**, and the following descriptions have been added to the main text or supplementary information.

Since chemically synthesized gold nanoparticles tend to aggregate under freezing conditions but cryogenic storage is necessary to conserve the bioactivity of Ausome, therefore we examined the stability of Ausome after repeated freeze-thaw cycles or long-term freezing at -80°C . No sediment was apparent in the thawed solutions and the TEM images exhibit monodispersed nanoparticles. DLS detected sizes and surface charges consistent with the fresh prepared Ausome further confirmed its stability under these storage conditions (**Supplementary Fig. 5a-c**). We also evaluated the influence of laser irradiation on the physiochemical features of Ausome. After exposing to laser for 30 min, Ausome was assessed using TEM and DLS, and little difference in morphology, diameter and zeta potential was observed when compared to fresh prepared Ausome (**Supplementary Fig. 5a-c**), revealing high laser stability of Ausome.

Supplementary Figure 5. Stability of Ausome under different storage conditions and laser irradiation. Representative TEM images (a), size distributions (b), and surface zeta potentials (c) of Ausome after repeated freezing and thawing (three cycles in one-week intervals, left), long time freezing (-80°C for 6 months) or laser irradiation ($1.5\text{ W}/\text{cm}^2$) for 30 min. Scale bars, 200 nm.

6. The resolution and quality of some of the figures should be checked carefully and improved, the fonts in figures are not clear and need to be enlarged.

Our response 6: Thanks for pointing out this issue, and we have reuploaded some IF, IHC and SEM images, and enlarged the fonts in Figures.

7. The herein described “Ausome” reminds me of the recently reported bacteria-mimetic gold nanoparticles, which are gold nanoparticles coated with OMVs (Science Advances, 2022, 8, eabn1805). These bacteria-mimetic nanoparticles play exactly the same role, with photothermal properties and eliciting immune response. The authors are suggested to cite and discuss about this work, and comment on the differences. In a similar sense, the reviewer believes that the “Ausome” is likely also transported by macrophages to the tumor, relevant discussions should be added.

Our response 7: We appreciate the reviewer for kindly sharing this interesting study. The design about hitchhiking phagocytic immune cells to inflammatory tumor tissue is really a brilliant idea for tumor-targeted delivery of nanomaterials with high

immunogenicity; the as planned aggregation of nanoparticles after internalizing to phagocytic immune cells not only avoids the leakage of cargoes, but also make the maximum absorption peak of the materials shift to longer wavelength. The aggregation promotes the photothermal conversion efficiency of the materials under laser with longer wavelength and allows for the using laser of with deeper penetration depth to elicit PTT. Gao's work reminds us an additional mechanism of the *in vivo* transport of Ausome, which is similar to the bacteria-mimetic nanoparticles studied in Gao's work. We have performed additional experiments and observed significant internalization of Ausome in macrophages. Moreover, Gao's study inspired us to further optimize the Ausome based antitumor therapy, based on the cell-based targeting delivery strategies and controlled aggregation of gold nanoparticles. We have included the discussion about the cell-based transporting of Ausome and the inspiration from Gao's work in the revised manuscript. The new data have been included in **Supplementary Figure 6c, d**, and the following descriptions have been added to the main text or supplementary information.

We then investigated interactions between Ausome and macrophages, one of the critical components of the first line of immune systems and respond quickly to exogenous stimuli; and macrophages can also be a hitchhike for the *in vivo* transport of Ausome³⁰. The bright field images exhibited large amount of intracellular gold nanoparticles after incubating with Ausome for 12 h. At the same time, AuNPs treated bone-marrow macrophages (BMDMs) showed little uptake of the nanoparticles (**Supplementary Fig. 6c**). ICP-MS further quantified the internalized Au, approximately 30% internalized Ausome and about 5% internalized AuNPs were measured, respectively (**Supplementary Fig. 6d**).

Supplementary Figure 6. Recognition and internalization of Ausome by innate immune cells. **c**, Bright field images of BMDMs after incubating with chemically synthesized AuNP or Ausome for 12 h. **d**, ICP-MS-quantified percentage of intracellular Au after incubation of BMDMs with Ausome for 12 h (n = 3 biological

independent samples). The numerical data in **(a, b, d)** are presented as the mean \pm s.d. * $p < 0.05$, ** $p < 0.01$; significant differences were analyzed by one-way ANOVA followed by the Bonferroni multiple comparison test **(a, b)** or student's t -test **(d)**.

Through genetic engineering or chemical conjugation, Ausome could be modified with specific targeting ligands, to promote its accumulation in tumor tissue, thus enhancing the ultimate anti-tumor therapeutic efficacy and optimizing the laser irradiation conditions, such as lower laser power density. Gao et al. recently reported a strategy of delivering *E. coli* OMV-coated gold nanoparticles to inflammatory tumor tissues by *in vivo* hitchhiking phagocytic immune cells, with ingenious design of intracellular self-assembly to minimize exocytosis of nanoparticles from immune cells³⁰. Gao's study exhibits another alternative for tumor targeted transport of Ausome, of which the clever design of gold nanoparticles aggregation could also lead to a red-shift of the maximum absorption, therefore, a longer wavelength laser with increased penetration depth could be used to inspire the photothermal effect of Ausome.

REVIEWERS' COMMENTS

Reviewer #1 (Remarks to the Author):

Review of NCOMMS-22-42773A

This revised manuscript uses an E Coli extraction coated gold-nanoparticle as an immunodulator that can induce thermal immunogenic cell death on the tumor. This is a novel approach in the preclinical space. The authors have addressed many of the immunological concerns to improve the rigor of their platform, particularly in Fig 2, 3, 4, R1, and Supp Fig 9. In terms of innovation, this is high. The authors addressed key MOA issues from the last submission. There are still concerns on translation, but overall this is a much improved proof of concept manuscript.

Reviewer #2 (Remarks to the Author):

The authors have addressed the problem very well, and the manuscript can be accepted in the present form.

Reviewer #3 (Remarks to the Author):

I have spent a few days to thoroughly looked into the revised manuscript as well as the response letter. The authors have taken great efforts to address all of the concerns and previous comments, I feel that the quality of the current manuscript has significantly improved. Thus I am happy to recommend acceptance of this work for publication.

POINT-BY-POINT RESPONSE TO THE REVIEWER'S COMMENTS:

We thank the reviewers for their constructive and valuable comments. Below, the reviewers' comments are shown in blue bold font and our responses are in standard typeface.

REVIEWER COMMENTS:

Reviewer #1: This revised manuscript uses an E Coli extraction coated gold-nanoparticle as an immunodulator that can induce thermal immunogenic cell death on the tumor. This is a novel approach in the preclinical space. The authors have addressed many of the immunological concerns to improve the rigor of their platform, particularly in Fig 2, 3, 4, R1, and Supp Fig 9. In terms of innovation, this is high. The authors addressed key MOA issues from the last submission. There are still concerns on translation, but overall this is a much improved proof of concept manuscript.

Our response: Thank you for your feedback on our revised manuscript. We appreciate your positive comments regarding our novel approach of using an E.coli extraction coated gold-nanoparticle as an immunomodulator to induce thermal immunogenic cell death on tumors. We are glad that you have recognized the improvements we made to address immunological concerns and enhance the rigor of our platform. We understand that there are still concerns regarding translation, which we will take into consideration for future studies.

Reviewer #2: The authors have addressed the problem very well, and the manuscript can be accepted in the present form.

Our response: We greatly appreciate your recognition of our efforts in addressing the issues raised during the review process.

Reviewer #3: I have spent a few days to thoroughly looked into the revised manuscript as well as the response letter. The authors have taken great efforts to address all of the concerns and previous comments, I feel that the quality of the current manuscript has

significantly improved. Thus I am happy to recommend acceptance of this work for publication.

Our response: Thank you for taking the time to thoroughly evaluate the revised manuscript and response letter. We greatly appreciate your efforts in assessing the improvements made to address the concerns and comments raised during the review process.